# DMNet: Self-comparison Driven Model for Subject-independent Seizure Detection

**Shihao Tu**
Zhejiang University
`shihao.tu@zju.edu.cn`

**Linfeng Cao**
The Ohio State University
`cao.1378@osu.edu`

**Daoze Zhang**
Zhejiang University
`zhangdz@zju.edu.cn`

**Junru Chen**
Zhejiang University
`jrchen_cali@zju.edu.cn`

**Lvbin Ma**
Zhejiang Huayun
Information Technology Co. Ltd
`gmmmfly@163.com`

**Yin Zhang**
Zhejiang University
`yinzh@zju.edu.cn`

**Yang Yang**[†]
Zhejiang University
`yangya@zju.edu.cn`

## Abstract

Automated seizure detection (ASD) using intracranial electroencephalography (iEEG) is critical for effective epilepsy treatment. However, the significant domain shift of iEEG signals across subjects poses a major challenge, limiting their applicability in real-world clinical scenarios. In this paper, we address this issue by analyzing the primary cause behind the failure of existing iEEG models for *subject-independent* seizure detection, and identify a critical universal seizure pattern: seizure events consistently exhibit higher average amplitude compared to adjacent normal events. To mitigate the domain shifts and preserve the universal seizure patterns, we propose a novel *self-comparison* mechanism. This mechanism effectively aligns iEEG signals across subjects and time intervals. Based on these findings, we propose *Difference Matrix-based Neural Network* (DMNet), a subject-independent seizure detection model, which leverages self-comparison based on two constructed (*contextual*, *channel-level*) references to mitigate shifts of iEEG, and utilize a simple yet effective *difference matrix* to encode the universal seizure patterns. Extensive experiments show that DMNet significantly outperforms previous SOTAs while maintaining high efficiency on a real-world clinical dataset that we collected, as well as two public datasets for subject-independent seizure detection. Moreover, the visualization results demonstrate that the generated difference matrix can effectively capture the seizure activity changes throughout the seizure evolution process. Additionally, we deploy our method in an online diagnosis system to illustrate its effectiveness in real clinical applications.

## 1 Introduction

*Epilepsy*, a chronic neurological disorder, affects more than 65 million people around the world. Up to 70% of people with epilepsy can be free from seizure only if the seizure onset zone (SOZ) can be located and surgically removed [26]. To diagnose epilepsy, doctors rely on the assessment of

---

[†]Corresponding authors.

38th Conference on Neural Information Processing Systems (NeurIPS 2024).

electrical activities that reflect the state and function of the subject's brain. Electroencephalography (EEG) is a widely employed and cost-effective method to record these electrical activities by placing sensors on the scalp. However, as a non-invasive method, it is unable to accurately locate SOZ in the deep structures of the brain.

Nowadays, iEEG is widely employed to identify and locate SOZ. *Stereo-EEG*, one representative iEEG technique, involves the deep implantation of electrodes within the brain to record electrical activities. These electrodes contain multiple recording contacts, called *channels*, and are placed across different regions of the brain, which provide stereoscopic recordings of the brain from both cortical and subcortical structures simultaneously [30]. This fully developed technique has been proven to be both effective and safe [5].

Given a substantial volume of iEEG data, we present a pipeline tailored to real-world application scenarios for automated seizure detection (ASD), as depicted in Fig. 1 (g). Firstly, the ASD model is trained using data from accessible subjects. Subsequently, the trained model is applied to identify seizures in iEEG recordings from previously unseen subjects. Doctors can refer to the prediction results, allowing for a more accurate diagnosis and facilitating more effective treatment decisions.

However, most existing ASD methods are built on non-invasive EEG with a *subject-specific* setting [38, 31] and *subject-independent* setting [2, 44, 11]. However, these EEG-based methods are prone to failure when applied to iEEG data. This is primarily due to the significantly higher complexity of iEEG signals compared to EEG signals. iEEG signals exhibit a greater level of intricacy as a result of the structural and functional disparities in brain neural activities. Furthermore, there are variations in the number and placement of invasive electrodes (Fig. 1 (a,b)), leading to significant domain shifts across different subjects. However, Existing methods that employ domain adversarial training for *subject-independent* seizure detection on iEEG signals may encounter negative transfer effects [23, 27]. Although [41] proposed a method that utilizes a series of intricate pre-training strategies to learn the general pattern across subjects, it lacks efficiency. Consequently, developing an effective and efficient subject-independent ASD method using iEEG is crucial for clinical diagnosis. Here, we discuss the primary challenges associated with iEEG in ASD.

**Challenges.** The factors mentioned above cause a significant domain shift between subjects, posing an open question regarding the generalization of *subject-independent* epilepsy seizure detection using iEEG. This issue gives rise to challenges at both the subject level and the channel level:

**(1)** How to capture the general distinguishable representation for normal and seizures between different subjects and time intervals? Due to individual differences that exist among subjects, the inherent properties of iEEG recordings, such as amplitude, frequency, and others, are personalized to each subject. Even within the same individual, brain activities vary over time.

**(2)** How to reduce the inconsistency of the seizure patterns of different channels? The channels exhibit diverse patterns due to the iEEG records of various regions of the brain, potentially leading to conflicting patterns between subjects. For example, normal and seizure waves are indistinguishable between subjects or channels. As shown in Fig. 1, the normal wave in (e) is difficult to distinguish from the seizure waves in (c) and (d), while the seizure event in (f) is often mistaken for the normal one. Therefore, the second challenge is to personalize the representations of brain activity independent of channels, allowing the model to adapt to different subjects.

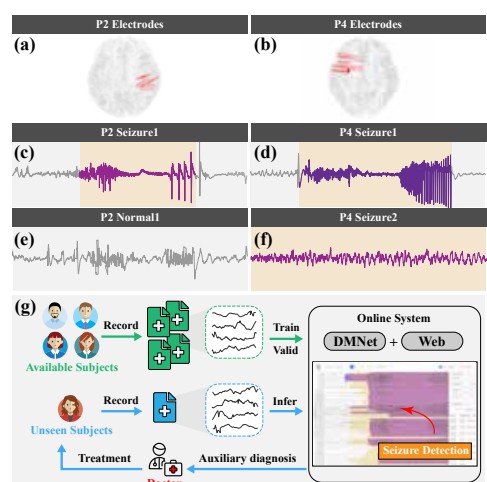

Figure 1: (a, b) Locations of iEEG depth electrode contacts (red circle) for subjects *P2* and *P4*. (c, d, e, f) Examples of seizure and normal iEEG recording activities of subjects *P2* and *P4*. (g) Application of our proposed method in real-world clinical scenarios.

**Solution.** To address these challenges, we first propose the mechanism of *self-comparison*: *comparing the target segment with adjacent normal segments*. Subsequently, we conduct a comprehensive observation study (Sec. 3) to demonstrate the effectiveness of *self-comparison*. Our findings indicate that *self-comparison* can obtain a general distinguishable representation of normal ones and seizures.

Drawing on these inspirations, we argue that the *self-comparison* mechanism is the key to easily and effectively capture subject-invariant patterns between subjects. To this end, we propose a novel model, namely ***D**ifference **M**atrix based neural **Net**work* (**DMNet**), for subject-independent seizure detection. Specifically, considering that different seizure events would present different neural activities (local bias) within different recording channels (global bias), we therefore introduce two reference objects (i.e., *contextual reference* and *channel-level reference*). These references ensure that we can capture both local and global dependencies within data, which are the primary contributors of distribution shift and can be effectively mitigated through the self-comparison mechanism. Subsequently, we utilize a simple yet effective *fully differencing operation* to generate the difference matrix, which compares the target segment with its reference objects for *self-comparison* implementation. To effectively extract semantics from the difference matrix, we design a difference matrix encoder based on convolutional neural network (CNN) blocks to obtain the final representation of the detection segment. Our primary contributions are listed as follows:

- We investigate the problem of subject-independent ASD based on the iEEG. Through comprehensive analysis, we identify the *self-comparison* mechanism as a simple yet effective way to capture the general representation.

- We propose a novel model named **DMNet** for subject-independent ASD. The *fully differencing operation* based on *contextual reference* and *channel-level reference* for self-comparison can mitigate the local and global biases among subjects and channels, improving the generalizability of learned representations.

- Extensive experiments on clinical and public iEEG datasets show *DMNet* outperforms existing SOTAs. Moreover, the generated difference matrix effectively captures seizure activity changes during the seizure evolution process. Furthermore, *DMNet* outperforms existing SOTAs while maintaining the high efficiency. Building on these strengths, we deploy our method in an online system, enhancing clinical applications by assisting medical professionals in the diagnosis of epilepsy and by facilitating the provision of more effective treatment options for patients.

## 2 Problem Formulation

In this work, the iEEG recording is regarded as a set of time series $\mathbf{X} = \{\boldsymbol{x}^{(i)}\}_{i=1}^{C}$, where $C$ refers to the total number of channels. Each $\boldsymbol{x}^{(i)} \in \mathbb{R}^{T}$ corresponds to a channel, and $T$ refers to the total timestamp. We sequentially test the channels one after another. Specifically, given *one channel* of iEEG time series $\boldsymbol{x}^{(i)} = \{x_1^{(i)}, \cdots, x_T^{(i)}\}$ from a subject, we first divide the original recording data into segments for detection. For simplicity, we omit the channel index $i$ in subsequent steps:

$$\{s_0, s_1, ..., s_{m-1}\} \tag{1}$$

where $s_k = \{x_{\ell \times k + 1}, \cdots, x_{\ell \times (k+1)}\}, 0 \leq k \leq m - 1$, $\ell$ is the number of timestamp for each segment, $m$ is the total number of segments. Each segment $s_k$ has a corresponding label $y_k \in \{0, 1\}$, indicating whether the segment contains a seizure event. In this work, our aim is to predict $y_k$ on each segment $s_k$ for different subjects.

We define our problem as an innovative study of *Domain Generalization* (DG) [45] in the context of epileptic diagnosis. In this study, we treat each subject's data as a domain, and our goal is to utilize the data of available labeled subjects (source domains) to train a model that can be directly adopted to the subjects with unseen data (target domains).

## 3 Empirical Analysis

In this section, we first analyze the primary cause behind the failure of existing models for subject-independent seizure detection of the iEEG and the reasons of occurrence. Then we explore the possibility of a domain-consistent seizure pattern existing within iEEG data, taking into account the

domain shift issue. Finally, we validate whether *self-comparison* mechanism can mitigate distribution shifts in iEEG data and capture the potential domain consistent seizure patterns.

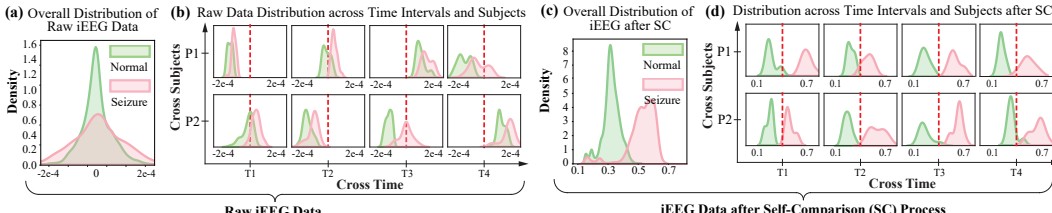

Figure 2: Observation results of the clinical iEEG dataset. (a) Overall distribution of raw iEEG signals (all subjects and channels), where normal and seizure events are indistinguishable. (b) Distribution of raw iEEG signals across different time intervals and subjects, where substantial domain shifts are evident in both distinct time intervals and among different subjects (red dashed line at 0 serves as reference line for domain shift). (c) Overall distribution of raw iEEG data after the *self-comparison* process. (d) Distribution of raw iEEG data across different time intervals and subjects after the *self-comparison* process. The *self-comparison* mechanism effectively mitigates distribution shifts across time intervals and subjects, thus enhancing the model's ability to distinguish between seizure and normal events.

## 3.1 iEEG Domain Shift Issue and Domain Consistent Seizure Pattern

Most previous studies [10, 16] assert that vanilla detection models trained independently by each subject are prone to failure when applied to other subjects. To analyze the direct cause of failure, we merge raw iEEG signals from all subjects and channels in a clinical iEEG dataset (details in App. C), and analyze the distributions of seizure and normal events. As depicted in Fig. 2 (a), the results reveal nearly identical means and close variances for both seizure and normal signals in the merged data. This similarity leads to the indistinguishability between normal and seizure signals across subjects, which becomes a direct factor to the failure of subject-dependent models.

For a detailed analysis, we partition the iEEG signals into multiple intervals, each comprising 250 timestamps, for each subject. We then compute the distributions of seizure and normal events within each interval. Through empirical analysis, we observe significant domain shifts both across different subjects (inter-subject) and different time intervals for the same subject (intra-subject). Fig. 2 (b) presents the normal and seizure distributions within four intervals randomly sampled from subjects P1 and P2, where the distribution patterns of normal and seizure exhibit substantial variation across different time intervals and subjects. This observation aligns with findings from previous studies in neural science and medicine, which have consistently reported that brain signals exhibit high variability between subjects and sessions due to inherent background neural activities [33], seizure patterns [14], electrode locations [9], etc.

Despite the pronounced domain shift observed in iEEG signals across subjects and time intervals, there is a notable commonality among subjects. Specifically, seizure events consistently demonstrate a higher average amplitude in the frequency domain compared to their background signal (adjacent normal events), indicating more intense neural oscillatory activities. This finding is consistent with previous studies in the field [34, 21].

## 3.2 Self-Comparison can Help

Based on the commonality above, we propose a novel *self-comparison* mechanism, which compares the target segment with its adjacent normal segment, to mitigate domain shifts between subjects and time intervals. To verify the effectiveness of this mechanism, we conduct an empirical study. First, for each subject, we partition the contiguous iEEG data into small segments. Considering that the spectral signal of brain data can effectively track transient changes before and during seizures [6], we proceed to transform these segments into the frequency domain using Discrete Fourier Transform (DFT). Subsequently, we calculate the spectrum differences by subtracting the spectrum of the target segment from those of adjacent segments on both sides of the target segment. We then sum up the absolute values of these differences, generating a single value $D$ for each target segment.

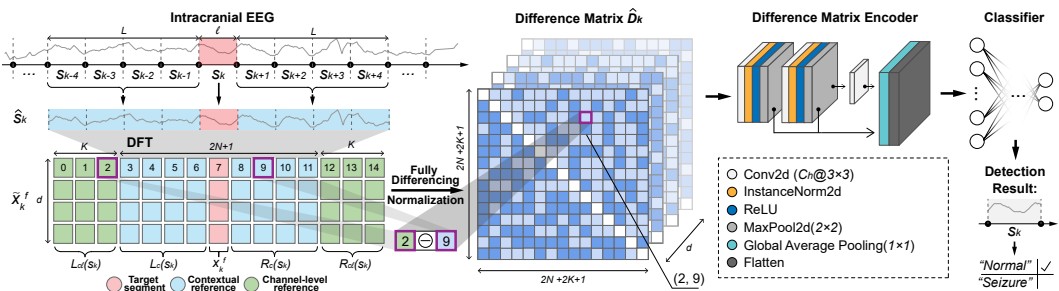

Figure 3: Overview of the proposed **DMNet**.

The overall distributions of $D$ values for all seizure and normal segments are depicted in Figure 3 (c). Notably, the distribution of the normal and seizure segments becomes distinctly separated after the adoption of the self-comparison mechanism. Additionally, we analyze the distribution of $D$ values for target segments within the same four intervals from P1 and P2, which is discussed in Section 3.1. The results are shown in Figure 3 (d). It is evident that the distribution of normal or seizures segments is well aligned across different subjects and time intervals, and the patterns of normal and seizure become more distinguishable. These results signify that our proposed self-comparison mechanism effectively mitigates the domain shift issue and preserves a domain consistent and distinguishable representation for normal and seizure segments.

## 4 Methodology

Inspired by the above observations, we propose a subject-independent seizure detection framework called **DMNet**, which leverages self-comparison to alleviate the distribution shift and preserve the domain consistency while distinguishing seizure patterns.

**Overview.** The overall framework is illustrated in Fig. 3. First, for each detection segment $s_k$ ▉, we construct two reference segments (*contextual reference* ▉: $L_c(s_k)$ / $R_c(s_k)$, *channel-level reference* ▉: $L_{c\ell}(s_k)$ / $R_{c\ell}(s_k)$) to compare with the target segment (Fig. 3, left). Then we use a simple yet effective *fully differencing* operation with signed-min-max normalization for self-comparison implementation, and the compared information is encoded by a *difference matrix* (DM) (Fig. 3, middle). Then we employ a CNN-based difference matrix encoder to learn the latent representation of DM, and use a classifier for seizure detection (Fig. 3, right).

### 4.1 References for Self-Comparison

**Contextual Reference.** The structural and functional differences in brain neural activity lead to distribution shifts between subjects, and even across different time intervals within the same subject. Based on the insights from observation studies in Section 3.2, we propose to compare the seizure wave with its adjacent contexts for the identification of seizure activities, as it significantly mitigates the domain shifts between subjects and preserves a domain-consistent and distinguishable pattern of seizure events.

However, capturing the long dependencies between seizure events and their contexts is challenging with a single contextual segment. This is because a typical epileptic seizure phase consists of pre-seizure aura, seizure onset, and post-seizure periods, with varying duration (ranging from seconds to minutes or longer in the case of status epilepticus) [12]. To this end, we first extract $2 \times N$ temporal segments $\{s_i\}_{k-N \le i \le k-1}$ and $\{s_i\}_{k+1 \le i \le k+N}$ from both sides of $s_k$ (each contains $L = N \times \ell$ timestamps in total, with each segment consisting of $\ell$ timestamps), ensuring that the context contain abundant normal information for comparison. Subsequently, we apply DFT to convert these segments into the frequency domain representation $\in \mathbb{R}^d$. Finally, the frequency domain segments obtained from the left and right sides are referred as the contextual references of $s_k$, denoted as $L_c(s_k) \in \mathbb{R}^{N \times d}$ and $R_c(s_k) \in \mathbb{R}^{N \times d}$ (▉) respectively.

**Channel-level Reference.** Different physiological brain regions have variations in neural activities [37], leading to distribution shifts between channels, even within the same subject. Although

*contextual* reference can reduce local bias, it fails to address global bias. Moreover, for prolonged epileptic seizures, solely considering the adjacent contextual reference segments may not provide sufficient normal information as background for comparison. To address these issues, we introduce *channel-level* reference as representative features of channels. The aim is to personalize channels, alleviate global bias, and provide comprehensive global background information of normal events.

Specifically, we adopt the K-Means [24] algorithm to identify the most representative patterns within a channel. First, we divide the entire time series of each channel into segments with length $\ell$. Then, similar with contextual reference, we use DFT to obtain the frequency domain representation $\in \mathbb{R}^d$ of all segments. Next, the K-Means clustering algorithm is applied to group all frequency domain representations into $K$ clusters. Finally, we arrange the clusters in descending order according to the number of elements they contain (denoted as $C_1, C_2, \cdots, C_K$, where $|C_1| \geq |C_2| \geq \cdots \geq |C_K|$). The arrangement indicates that the higher the index of the cluster, the lower the frequency of occurrence of the corresponding general pattern in the respective channel.

To obtain the general patterns of the channel, we use the centroid of each cluster, resulting in the final representation denoted as $\mu_k \in \mathbb{R}^d, k = 1, ..., K$. These $\mu_k$ values are then concatenated to construct the left side channel-level reference $L_{c\ell}(s_k) \in \mathbb{R}^{K \times d}$. Empirically, the right side $R_{c\ell}(s_k)$ is formed by reversing the order of $L_{c\ell}(s_k)$.

## 4.2 Difference Matrix

In this subsection, we present the self-comparison implementation based on the constructed references and target segments through a simple yet effective approach of *fully differencing operation*. The comparison information is encoded using a *difference matrix* (DM).

We first obtain the frequency domain representation $x_k^f$ (▦) of target segment $s_k$ via DFT, and then concatenate $x_k^f$ with the constructed contextual (▦) and channel-level (▦) references to form the augmented segment $\tilde{x}_k^f$ (as shown in Figure 3, bottom-left):

$$\tilde{x}_k^f = L_{c\ell}(s_k) \, ||L_c(s_k)|| \, x_k^f \, ||R_c(s_k)|| \, R_{c\ell}(s_k), \tag{2}$$

where $||$ is a concatenate operation and $\tilde{x}_k^f \in \mathbb{R}^{(2N+2K+1) \times d}$.

**Fully Differencing Operation.** To implement self-comparison of target segment and references, we introduce a fully differencing operation. Unlike traditional first-order differencing [13] that considers only adjacent points, the fully differencing operation makes a pairwise comparison across all segments in $\tilde{x}_k^f$ (described in Equation 3 and Fig. 3, middle), which can more effectively capture the essential seizure patterns of brain activities. Moreover, since there are inherent scale differences in the vanilla difference matrix $\mathbf{D}_k$ caused by varying magnitudes between seizure and normal iEEG signals across subjects, channels and time intervals, min-max normalization is adopted to address the scale difference issue. After these two operations, a synthetic difference matrix $\hat{\mathbf{D}}_k$ can be obtained:

$$\mathbf{D}_k[i, j] = \tilde{x}_k^f[i] - \tilde{x}_k^f[j], 1 \leq i, j \leq 2N + 2K + 1, \tag{3}$$

$$\hat{\mathbf{D}}_k = \text{Min-Max-Norm}(\mathbf{D}_k) \in \mathbb{R}^{(2N+2K+1) \times (2N+2K+1) \times d} \tag{4}$$

The generated difference matrix $\hat{\mathbf{D}}_k$ contains rich semantic information about the evolution of seizures. We provide a more detailed discussion on each component of the difference matrix $\hat{\mathbf{D}}_k$ and the corresponding semantic properties in App B.

**Difference Matrix Encoder.** To well capture and learn these essential differences, we adopt the CNNs as the DM encoder (Fig. 3,*right*), which have been proven to be powerful in learning representations from 2D matrices [35, 42]. The output of the CNNs will be concatenated as a representation $\hat{\mathbf{Z}}$ of the DM. Finally, $\hat{\mathbf{Z}}$ will be fed into a linear classifier to obtain the detection result.

## 5 Experiments

In this section, we conduct extensive experiments on public and clinical iEEG datasets to address three primary research questions: **RQ1.** How does the proposed *DMNet* model perform in subject-independent seizure detection compared to other methods? **RQ2.** Do the proposed contextual

Table 1: Average performance of subject-independent seizure detection tasks on clinical & public datasets. The **v** indicates the first rank in a column and v indicates the second. The performance with standard deviation is given in App. G.

| Dataset / Model | Clinical | | | | MAYO | | | | FNUSA | | | |
|---|---|---|---|---|---|---|---|---|---|---|---|---|
| | Pre. | Rec. | F1 | F2 | Pre. | Rec. | F1 | F2 | Pre. | Rec. | F1 | F2 |
| SelfReg | 51.60 | 48.74 | 51.24 | 48.63 | 60.40 | 32.13 | 36.13 | 32.12 | 62.54 | 48.19 | 49.20 | 47.73 |
| GroupDRO | 47.60 | 44.74 | 45.15 | 46.33 | 48.31 | 35.00 | 27.82 | 28.04 | 53.48 | 71.44 | 60.47 | 66.38 |
| MTL | 20.46 | 52.33 | 28.59 | 39.13 | 46.87 | 22.08 | 15.68 | 16.31 | 60.04 | 52.64 | 53.90 | 52.83 |
| CORAL | 38.70 | 49.20 | 42.01 | 47.66 | 62.17 | 29.86 | 20.41 | 22.01 | **65.13** | 53.23 | 55.93 | 53.88 |
| CDANN | 33.43 | 40.41 | 35.72 | 37.58 | 36.79 | 79.55 | 45.49 | 56.60 | 64.37 | 54.85 | 54.35 | 53.86 |
| SD | 18.69 | 54.40 | 28.78 | 40.81 | 47.73 | 55.59 | 46.97 | 50.35 | 56.99 | 57.97 | 55.42 | 56.45 |
| IB-IRM | 29.19 | 49.75 | 37.91 | 42.64 | 47.57 | 57.17 | 46.86 | 50.71 | 54.22 | 63.26 | 55.96 | 59.47 |
| VREx | 44.80 | 32.45 | 36.33 | 35.34 | 51.21 | 59.95 | 51.19 | 54.85 | 54.74 | 60.15 | 54.64 | 57.12 |
| IB-ERM | 40.30 | 37.40 | 37.59 | 37.19 | 46.29 | 57.36 | 47.21 | 51.44 | 54.64 | 55.26 | 52.68 | 53.70 |
| TRM | 34.03 | 42.74 | 38.93 | 41.58 | 47.55 | 58.97 | 43.96 | 47.87 | 60.68 | 58.46 | 56.00 | 56.74 |
| Abou-Abbas et al. | 43.24 | 45.84 | 43.15 | 46.95 | 48.47 | 51.40 | 50.69 | 48.86 | 49.83 | 56.90 | 52.33 | 57.53 |
| Zhao et al. | 30.17 | 49.44 | 36.16 | 42.65 | 37.07 | 56.06 | 26.17 | 38.48 | 41.64 | 44.20 | 40.62 | 42.12 |
| Dissanayake et al. | 40.12 | 39.29 | 38.30 | 40.82 | 50.39 | 68.99 | 57.69 | 64.82 | 63.85 | 76.01 | 63.75 | 67.94 |
| SICR | 46.27 | 43.91 | 45.65 | 43.86 | **79.01** | 63.29 | 69.88 | 66.17 | 63.78 | 66.77 | 64.25 | 65.10 |
| SEEGNet | 44.89 | 47.70 | 46.25 | 47.11 | 71.82 | 60.50 | 64.87 | 63.15 | 62.23 | 72.35 | 66.81 | 68.12 |
| PPi | 51.72 | 49.70 | 49.78 | 51.12 | 74.49 | 70.21 | 72.28 | 71.02 | 59.53 | 75.42 | 65.83 | 71.59 |
| DMNet | **59.58** | **55.24** | **54.49** | **55.93** | 68.82 | **90.06** | **73.08** | **81.54** | 62.30 | **85.39** | **67.80** | **75.15** |
| DMNet w/o $L_{c\ell}$ | 48.25 | 53.30 | 49.62 | 51.20 | 47.10 | 89.15 | 62.63 | 76.43 | 52.57 | 78.49 | 60.99 | 70.75 |
| DMNet w/o $L_c$ | 51.39 | 47.43 | 47.32 | 47.15 | 58.34 | 76.73 | 64.28 | 71.79 | 49.28 | 73.48 | 58.12 | 65.23 |
| DMNet w/o **DM** | 43.58 | 45.79 | 46.72 | 43.42 | 49.67 | 71.54 | 60.63 | 66.57 | 46.98 | 66.89 | 56.21 | 62.18 |

reference, channel-level reference and difference matrix contribute to seizure detection? **RQ3.** How does the difference matrix reflect seizure activity changes during seizure evolution process?

## 5.1 Experimental Setup

**Datasets.** To evaluate the performance of our *DMNet* model, we conduct experiments on both the public benchmark dataset, which includes MAYO and FNUSA [25], and the private clinical dataset (details refer to App. C).

**Evaluation Metrics.** For fair comparison, we use precision, recall, F1-score and F2-score as evaluation metrics. Typically, F2-score is particularly emphasized in practical clinical studies [15, 43] since overlooking any seizure can be costly in terms of diagnosis. Therefore, in our study, the F2-score serves as the primary metric for comparison.

**Settings.** To conduct the experiment under the domain generalization settings, we divide the subjects in the datasets into multiple groups and assign different groups as source and target domains for model training, validation and testing. A more detailed description of experimental setup that includes *DMNet* hyperparameters and setup on clinical and public datasets, can be found in App. D.

**Baselines.** We compare the proposed *DMNet* with state-of-the-art subject-independent seizure detection algorithms for both iEEG-based methods and EEG-based methods. For iEEG-based methods, we compared against SICR [17], SEEG-Net [39], and PPi [41]. For EEG-based methods, we compared against Abou-Abbas et al. [2], Zhao et al. [44], and Dissanayake et al. [11]. Additionally, we compare the performance with the domain generalization (DG) algorithms in other areas like Self-Reg [18], GroupDRO [32], MTL [7], CORAL [36], CDANN [22], SD [28], IB-IRM [3], VREx [19], IB-ERM [4], TRM [40]. More details of the baselines are shown in App. A.

## 5.2 Overall Performance Comparison (RQ1)

The overall performance of our proposed model *DMNet* and other baselines for subject-independent seizure detection are presented in Tab. 1. From the results, we can see that our proposed *DMNet* significantly outperforms other SOTA subject-independent seizure detection algorithms and the latest domain generalization methods, with an average improvement of $9.41\%$, $14.81\%$ and $4.97\%$ in terms of F2 score on clinical and public datasets (MAYO and FNUSA) respectively. These results highlight the superior generalization ability of *DMNet*. Compared to DG baselines, our model exhibits a

substantial margin in all evaluation metrics, suggesting that general DG baselines fail to capture the diverse and evolving distribution patterns of iEEG over time. Furthermore, the EEG-based methods proposed by [2], [44] and [11] may not be adept at handling complex iEEG signals, which lead to subpar performance. Although the iEEG-based methods [41], SEEG-Net [39] and SICR [17] outperform most EEG-based methods and DG methods on F2-score, they still fall short of the performance compared to our model. This may be attributed to the fact that they do not explicitly capture the general pattern of difference between normal and seizure signals. In contrast, we utilize a difference matrix to achieve a distinguishable representation of normal and seizure.

## 5.3 Ablation Study of *DMNet* (RQ2)

To validate the contribution of each component of our proposed *DMNet*, we conduct ablation studies on key components (**w/o** means without and **w/** means with). The results of the ablation evaluation can be seen in Tab. 1, from which we can see full *DMNet* significantly outperforms other ablated models on F2-score. These results of the ablation experiment highlight the effectiveness of the following components:

**Channel-level Reference.** Removing channel-level reference causes poorer results compared to the full version. It indicates that utilizing the channel-level reference with representative characteristics of each channel for global dependencies modeling can improve the performance.

**Contextual Reference.** Removing contextual reference results in a noticeable performance drop. It illustrates that introducing the informative contextual reference for self-comparison to capture long-term dependencies and complete patterns of seizure improves the performances.

**Difference Matrix.** Removing DM leads to a significant drop in model performance, indicating the effectiveness of a fully differencing operation for the implementation of self-comparison.

## 5.4 Case Study (RQ3)

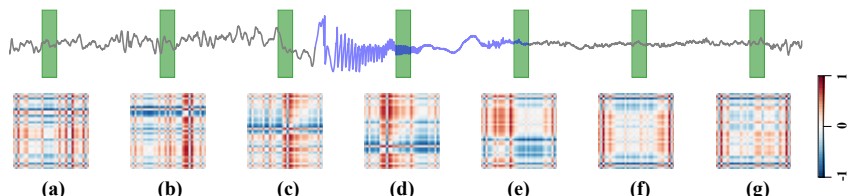

Figure 4: Case study.

To provide a more intuitive demonstration of *DMNet*, we present the visualization results of difference matrix throughout the seizure process in Fig. 4 (A full visualization can be found in App. E). The upper figure shows the raw brain signal containing a full seizure process, with the gray wave representing the normal signal and the purple wave representing the seizure. The green masked blocks indicate the segments for detection. Notably, there are clear distinctions between seizure and normal difference matrices during the seizure evolution. Segments being closer to seizure events show rougher difference matrices (e.g., segments c, d, and e), while those further away appear smoother (e.g., segments a, b, f, and g). This case clearly illustrates how the difference matrix captures seizure activity changes and demonstrates the effectiveness of *DMNet*.

## 5.5 Hyperparameters Analysis of *DMNet*

**Number of Segments $N$.** As described in Sec. 4.1, we utilize $2 \times N$ segments to form the contextual reference. Increasing the value of $N$ results in a longer contextual reference, indicating the inclusion of a greater amount of contextual information. As depicted in Fig. 5(a), the evaluation scores generally increase as $N$ increases from 8 to 12. This trend is attributed to the fact that a longer contextual reference can provide more comprehensive information throughout the entire seizure phase.

**Segment Length $\ell$.** We investigate the effects of segment length by varying the segment length $\ell$ of contextual reference. The performance of *DMNet* with different segment lengths $(100, 150, 200, 250, 300)$ is shown in Fig. 5(b). As $\ell$ increases, the model precision decreases.

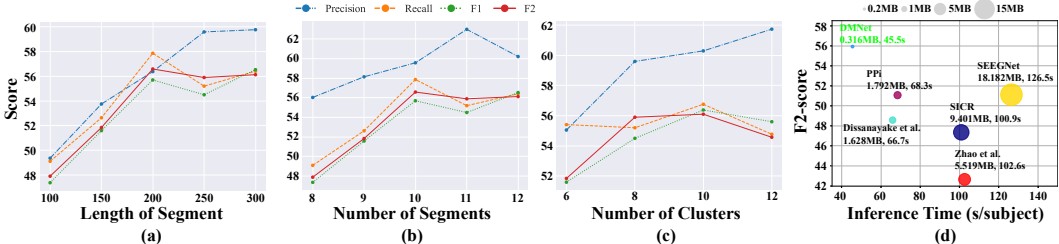

Figure 5: Hyperparameters Anslysis: (a, b, c) and Computational Efficiency Analysis (d).

In contrast, recall, F1 score and F2-score initially increase when $\ell$ ranges from 100 to 200 but then decline. This suggests that shorter segment lengths ($\ell$) fail to provide representative semantic information, preventing the model from capturing long-term dependencies in the brain signal. Conversely, excessively long segment lengths can result in coarse-grained temporal representations, leading to the loss of fine-grained patterns and details.

**Number of Clusters $K$.** We vary the number of clusters $K$ in channel-level reference, which controls the number of generated channel representative features of a specific channel. As we can see in Fig. 5(c), the evaluation metrics (recall, F1 score, and F2-score) demonstrate an initial increase followed by a decrease trend as $K$ varies from 6 to 12. However, precision shows an upward trend. This indicates that introducing global information can reduce false positive samples but it also affects recall. Therefore, it is necessary to consider trade-offs when selecting the value of $K$.

## 5.6 Generalization Ability Analysis

To further assess the generalization capability of DMNet on a broader range of subjects with greater heterogeneity, we evaluated the model on data of 179 previously unseen subjects from the large TUSZ EEG dataset [1]. Additional details about this evaluation study are provided in App.F.

## 5.7 Application Scenario

**Model Efficiency.** We compare the model efficiency of *DMNet* and several benchmark models in terms of parameter count and average inference time per patient file. As shown in Fig. 5 (d), the parameter count of *DMNet* is only 19.4% of the model with the smallest parameter count, but its performance is 109.4% of the best-performing model. Additionally, *DMNet* has an average inference time of 45.5 seconds per subject file, which is also the fastest among all models. Overall, *DMNet* demonstrates significant advantages in both parameter count and inference time. Therefore, *DMNet* is an ideal choice, providing a reliable and high-performance solution for real application scenario.

**Online Deployment.** *DMNet* has been deployed on an online system (Fig.6), which illustrates the effectiveness of our method in a real clinical application. This system serves as an auxiliary tool for expert doctors, significantly enhancing the accuracy and efficiency of the diagnostic process. The system comprises two important pages: the overview page and the detail page. The overview page (Fig.6 (*top*)) offers a comprehensive view of the 12-hour patient file. Each square on the page represents a 1-minute iEEG signal segment and is color coded to indicate various states, including no epileptic waves (*gray*), correct predictions (*green*), incorrect predictions (*blue*), and missing predictions (*red*) made by our model. By clicking on a square, doctors can access the detail page (Fig.6 (*bottom*)), where they can change the presented time period using the top toolbar. The page includes a data operation panel and seizure events displayed on the right

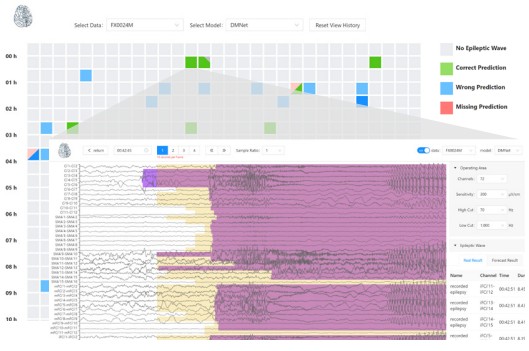

Figure 6: Online auxiliary diagnosis system.

side. In the center of the page, the purple section represents the true seizure annotations provided by doctors, while the yellow section showcases our model's predictions. As depicted in the figure, the predictions of our model align well with the actual seizures.

## 6    Related Work

**Epileptic Seizure Detection for EEG.** Data-driven methods for seizure detection have gained attention in clinical medicine. [38] combine CNN and SVM, [31] leverage frequency components and CNN. However, these methods are subject-specific and not suitable for real-world application. Several efforts have been made in the study of subject-independent methods in EEG. [2] proposed and investigates a subject-independent seizure detection model that uses stable EEG-based features obtained by comparing multiple feature selection methods. [44] proposed two subject independent deep learning architectures with different learning strategies that can learn a global function utilizing data from multiple subjects. [11] proposed IBA (Information Bottleneck Attribution), a subject-independent seizure detection model that utilizes multi-view information. It employs adversarial deep learning to learn seizure-specific feature representations directly from raw EEG data. However, these methods are based on non-invasive EEG recordings.

**Epileptic Seizure Detection for iEEG.** iEEG, through the placement of electrodes inside or on the surface of the brain, offers higher temporal and spatial resolution, enabling more accurate capture and analysis of brain activity in specific regions, including subtle changes in electrophysiological signals. Consequently, this has spurred research into epilepsy detection based on iEEG [26]. There are several seizure detection methods being designed in subject-independent settings, which are more applicable to real-world scenarios. SEEG-Net [39] and SICR [17] proposed adversarial training to learn subject-invariant features on iEEG recordings. However, their experiments were carried out on small, manually denoised datasets with a balanced positive-negative sample ratio, resulting in a significant data bias that deviates from real clinical requirements. Moreover, [41] proposed a method that utilizes a series of intricate pre-training strategies to learn general pattern cross subjects, which is lack of efficiency.

**General Domain Generalization.** Our work focuses on detecting seizure events in unseen subjects, which can be framed as the domain generalization (DG) problem [45]. Existing DG studies can be categorized into two groups: *Invariant representation based method* [36, 22, 4, 18, 3, 7] and *Learning strategy based method* [20, 32, 19, 28]. Invariant representation methods aim to learn domain-invariant representations. CORAL [36] aligns covariance in feature layers, enhancing the extraction of domain-invariant features. CDANN [22] introduces adversarial training to encourage robust and domain-invariant feature learning. IB-ERM [4] minimizes empirical risk across domains to improve generalization. SelfReg [18] uses self-supervised learning and contrastive loss to capture invariant information across domains. Learning strategy methods aim to enhance generalization capability through various learning strategies. GroupDRO [32] achieves higher worst-group accuracy by coupling robust optimization models with increased regularization. VREx [19] penalizes the variance of training risks to improve domain extrapolation. Despite prior efforts, DG problems in time series, like iEEG signals, continue to be a relatively unexplored area and poses considerable challenges due to the diverse and evolving distribution patterns with time.

## 7    Conclusion

In this paper, we present a novel seizure detection framework called DMNet. Our model addresses the challenge of generalization across different subjects by incorporating a self-comparison mechanism to capture the subject-invariant representation. Extensive experiments conducted on clinical intracranial EEG dataset and public dataset demonstrate the effectiveness of our model in subject-independent seizure detection tasks. Moreover, our generated difference matrix effectively captures seizure activity changes during the seizure evolution process, which is valuable for clinicians to better understand the seizure event and develop more effective treatment. Furthermore, *DMNet* outperforms existing SOTAs while maintaining the high efficiency. Based on these, we deploy our method in an online system, enhancing clinical applications by assisting physicians in the diagnosis of epilepsy and in offering optimal treatment strategies. We hope that this work will shed light on the development of a more robust subject-independent seizure detection system.

## Acknowledgments

This work was partially supported by National Natural Science Foundation of China (No. 62322606, No. 62441605).

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

# A  Baselines

Firstly, we compare our model to other iEEG-based subject-independent epilepsy detection models. The Details of these baseline models are given here:

- SICR [17]: a framework that learns class-relevant and subject- invariant feature representations, which shows a promising performance in non-invasive brain-computer interface.

- SEEG-Net [39]: a model that can address the problems of sample imbalance, cross-subject domain shift, and poor interpretability and realizes high-sensitivity SEEG pathological activity detection. The source code of SEEG-Net is not released, so we implement it by ourselves to conduct the experiments.

- PPi [39]: proposed a method that utilizes a series of pre-training strategies to extract rich information from iEEG data while preserving the unique characteristics between brain signals recorded from different brain areas.

Secondly, we also compare our model to several EEG-based subject-independent epilepsy detection models. The Details of these baseline models are given here:

- [2]: they propose and investigates a patient-independent seizure detection model that uses stable EEG-based features obtained by comparing multiple feature selection methods.

- [44]: they propose two subject independent deep learning architectures with different learning strategies that can learn a global function utilizing data from multiple subjects.

- [11]: they propose a subject-independent seizure detection model, called IBA (Information Bottleneck Attribution), that utilizes multi-view information. By employing adversarial deep learning, the model learns seizure-specific feature representations directly from raw EEG data.

Moreover, we offer more details on comparisons to other latest domain generalization methods utilized in this paper:

- CDANN [22]: an end-to-end conditional invariant deep DG approach by leveraging deep neural networks for domain-invariant representation learning.

- CORAL [36]: an unsupervised domain adaptation method that aligns the second-order statistics of the source and target distributions with a linear transformation.

- GroupDRO [32]: a model coupling group DRO models with increased regularization, where DRO allows to learn models that instead minimize the worst-case training loss over a set of groups.

- MTL [8]: a representative framework for DG, which augments the original feature space with the marginal distribution of feature vectors.

- SD [29]: a regularization method aimed at decoupling feature learning dynamics, improving accuracy and robustness in cases hindered by gradient starvation.

- SelfReg [18]: a regularization method for DG based on contrastive learning, self-supervised contrastive regularization.

- TRM [40]: a robust estimation criterion that is specifically geared towards optimizing transfer to new environments.

- VREx [19]: a penalty on the variance of training risks as a simpler variant based on a form of robust optimization over a perturbation set of extrapolated domains.

- IB-ERM [4]: a DG method that improve generalization via minimizes the empirical risk over multiple domains.

- IB-IRM [4]: a DG method that improve generalization via minimizes the invariant risk over multiple domains.

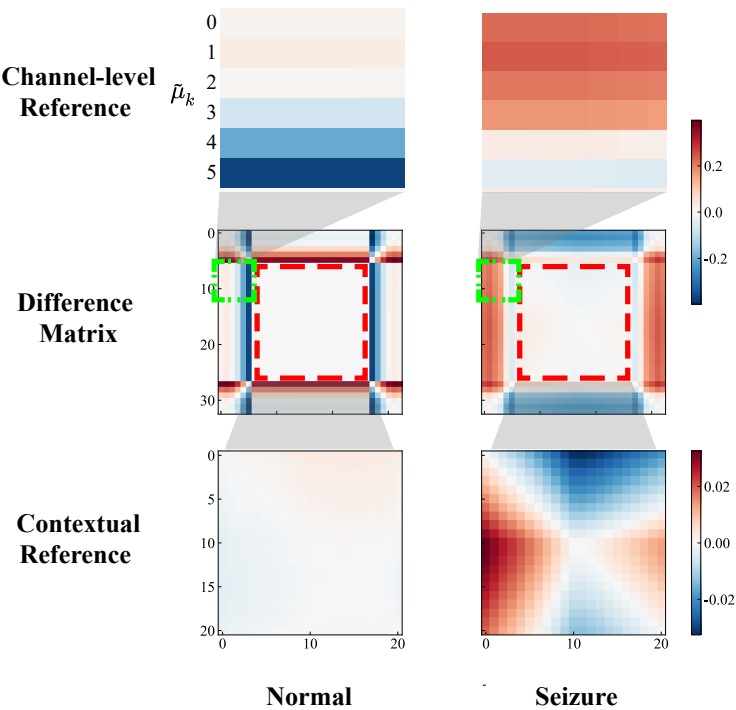

Figure 7: Visualization of difference matrix.

# B   Indepth Visualization of Difference Matrix

In this section, we present visualizations of the difference matrix (DM) constructed in Section 4.2, for providing an in-depth analysis of the DM properties. Initially, the difference matrix is constructed separately for normal and seizure segments of all subjects using the method mentioned in Section 4. Next, we average all difference matrices of normal and seizure segments to obtain the averaged 2-dimensional matrices. These two averaged 2D matrices are shown in the second row of Fig. 7. The green square with dashed line represents the difference between the channel-level reference and target segment $x_k^f$. A zoom-in visualization is provided in the first row of the Fig. 7. The left (normal) shows little difference of $x_k^f$ with most of the channel patterns (small cluster index), but significant difference with a small portion of channel patterns (large cluster index). Conversely, the right one (seizure) exhibits the opposite pattern, validating the effectiveness of channel-level reference. Furthermore, the red square with the dashed line represents the difference within $x_k^f$ (only containing contextual reference). The third row of the figure presents a zoom-in visualization, showing a smooth 2-dimensional matrix for normal event and a rough 2-dimensional matrix for seizure event. This demonstrates the effectiveness of introducing contextual reference.

# C   Datasets

To evaluate the performance of our model, we conduct extensive experiments on the following two datasets.

**Clinical Dataset.** The clinical dataset in our study is provided by a first-class hospital, and the intracranial EEG electrode implantation surgery and data collection are approved by the official ethics committee. For each subject, 4 to 10 invasive electrodes with 52 to 126 channels are implanted according to clinical needs to obtain brain signals. The dataset is quite massive due to the high frequency and multiple channels used to record intracranial EEG data, with more than 738 recording hours and 877.3GB total size. Data are labeled by professional neurologists at point level. Moreover, the positive sample (seizure event) ratio of a single subject in our dataset is around 0.003 on average, which is extremely imbalanced. More details can be found in Table 3.

**Public Dataset.** The public dataset MAYO and FNUSA [25] used in our paper, is collected from St. Anne's University Hospital (Brno, Czech Republic). This dataset is composed of intracranial EEG data collected in an awake resting state from 38 diagnosed subjects. Specifically, the dataset comprises a total of 348,300 segments. Each segment spans a duration of 3 seconds and consists of 15,000 data points, with a sampling frequency of 5000Hz. These segments are labeled into 4 categories, including physiological activity, pathological activity (seizure event), artifacts, and power line noise. We follow the setup in [39] of omitting invalid data from one subject while retaining the data from 30 subjects for analysis. More details can be found in Table 3.

Table 2: Basic hyperparameters of DMNet.

| Parameter | Clinical dataset | Public dataset |
|---|---|---|
| Length of Segment $\ell$ | 250 | 500 |
| Number of Segments $N$ | 10 | 7 |
| Number of Clusters $K$ | 8 | 8 |
| Base Filter Number $C_h$ | 64 | 8 |
| Batch Size | 24 | 24 |
| Optimizer | Adam | Adam |
| Learning Rate | $3 \times 10^{-4}$ | $3 \times 10^{-4}$ |
| Max Epoch | 20 | 20 |
| Valid Metric | F1-score | F1-score |

Table 3: Details information of the clinical and public dataset.

| Dataset | Subject ID | Time (hours) | Sample frequency(Hz) | #Electrodes | #Channels | Positive sample ratio | #Samples | gender |
|---|---|---|---|---|---|---|---|---|
| Clinical | P0 | 72 | 1000 | 10 | 126 | 0.003 | 8,113,760 | male |
| | P1 | 24 | 512 | 7 | 93 | 0.0002 | 4,176,200 | female |
| | P2 | 6 | 512 | 5 | 69 | 0.004 | 3,698,920 | male |
| | P3 | 21 | 1000 | 4 | 52 | 0.004 | 3,460,280 | female |
| | P4 | 114 | 1000 | 10 | 126 | 0.001 | 9,664,920 | female |
| | P5 | 36 | 512 | 5 | 63 | 0.002 | 3,818,240 | female |
| | P6 | 24 | 512 | 7 | 89 | 0.009 | 6,801,240 | female |
| | P7 | 36 | 1024 | 4 | 52 | 0.001 | 1,789,800 | male |
| MAYO | P0 | 2.2 | 5000 | - | - | 0 | 2648 | - |
| | P1 | 7.9 | 5000 | - | - | 0.1020 | 9536 | - |
| | P2 | 2.3 | 5000 | - | - | 2.2231 | 2788 | - |
| | P3 | 5.6 | 5000 | - | - | 0 | 6693 | - |
| | P4 | 2.4 | 5000 | - | - | 0 | 2853 | - |
| | P5 | 6.3 | 5000 | - | - | 0 | 7585 | - |
| | P6 | 10.7 | 5000 | - | - | 0 | 12873 | - |
| | P8 | 2.3 | 5000 | - | - | 1 | 2816 | - |
| | P9 | 0.6 | 5000 | - | - | 0 | 740 | - |
| | P14 | 3.2 | 5000 | - | - | 6.8795 | 3924 | - |
| | P16 | 3.2 | 5000 | - | - | 0 | 3876 | - |
| | P17 | 8.5 | 5000 | - | - | 0 | 10194 | - |
| | P18 | 4.0 | 5000 | - | - | 0 | 4826 | - |
| | P19 | 4.7 | 5000 | - | - | 0 | 5613 | - |
| | P20 | 2.3 | 5000 | - | - | 0 | 2702 | - |
| | P21 | 2.9 | 5000 | - | - | 59.1724 | 3490 | - |
| | P23 | 3.46 | 5000 | - | - | 1.96 | 4152 | - |
| FNUSA | P0 | 1.6 | 5000 | - | - | 1 | 1912 | - |
| | P1 | 10.3 | 5000 | - | - | 0.1341 | 12358 | - |
| | P2 | 6.7 | 5000 | - | - | 0.9985 | 8088 | - |
| | P3 | 7.1 | 5000 | - | - | 0 | 8463 | - |
| | P4 | 10.0 | 5000 | - | - | 0.1268 | 12038 | - |
| | P5 | 2.1 | 5000 | - | - | 0.6176 | 2516 | - |
| | P6 | 13.2 | 5000 | - | - | 0.4884 | 15843 | - |
| | P7 | 18.9 | 5000 | - | - | 0.0834 | 22774 | - |
| | P8 | 5.6 | 5000 | - | - | 1 | 6750 | - |
| | P9 | 9.7 | 5000 | - | - | 0.3675 | 11591 | - |
| | P10 | 8.6 | 5000 | - | - | 0.3953 | 10301 | - |
| | P11 | 39.3 | 5000 | - | - | 0.1631 | 47270 | - |
| | P12 | 16.4 | 5000 | - | - | 0.2708 | 19635 | - |

Table 4: Group information of clinical and public datasets.

| Dataset \ Group | A | B | C | D | E | F |
|---|---|---|---|---|---|---|
| Clinical | 0,7 | 5,6 | 2,4 | 1,3 | - | - |
| MAYO | 0,18,21 | 1,9,19 | 2,5,16 | 3,4,23 | 6,8 | 14,17,20 |
| FNUSA | 0,4 | 1,8 | 2,3,11 | 5,6 | 7,9 | 10,12 |

Table 5: Detailed information of dataset setup in the experiment.

| Dataset | Exp. id | Training | Validation | Test |
|---|---|---|---|---|
| Clinical | 0 | C+D | B | A |
| | 1 | B+D | C | A |
| | 2 | B+C | D | A |
| | 3 | C+D | A | B |
| | 4 | A+D | C | B |
| | 5 | A+C | D | B |
| | 6 | B+D | A | C |
| | 7 | A+D | B | C |
| | 8 | A+B | D | C |
| | 9 | B+C | A | D |
| | 10 | A+C | B | D |
| | 11 | A+B | C | D |
| Public | 0 | A+B+C+D | E | F |
| | 1 | A+B+C+F | D | E |
| | 2 | A+B+E+F | C | D |
| | 3 | A+D+E+F | B | C |
| | 4 | C+D+E+F | A | B |
| | 5 | B+C+D+E | F | A |

# D    Experimental Setup

All experiments were run on a Linux system with 2 CPUs (AMD EPYC 7H12 64-Core Processor) and 4 GPUs (NVIDIA GeForce RTX 3090).

**Setup on Clinical Dataset.** To conduct the experiment under domain generalization setting, we divide these subjects in a clinical dataset into 4 groups, each group contains 2 subjects (detailed grouping information is listed in Table 4). For model training and testing, we adopt a "2-1-1 setting" for the division of training, validation, and test sets. Specifically, we assign two groups to the training set and one group to the validation set, collectively forming the source domains. Additionally, we assign another group as the target domain for testing. For a comprehensive experiment, we conduct the experiments under 12 different grouping combinations (detailed settings in Table 5). To reduce computational load, we downsample the data to 250Hz. The length of the patch $\ell$ is set to 1 second (250 data points). More experimental details, including model and optimization settings, are listed in Table 2.

**Setup on Public Dataset.** For the public dataset, we employ the grouping strategy mentioned in [39]. Specifically, we divide these subjects into 6 groups. We adopt a "4-1-1 setting" for model training, validation, and testing, respectively: randomly choose 5 groups (4 testing groups and 1 validation group) as the source domain, while the other one group as the target domain for testing. For the public dataset, we conduct the experiments under 6 different grouping combinations (detailed setups in Table 5). Moreover, because the public dataset is sampled data, there is no complete channel data, so for DMNet with public data, we remove the channel-level reference. More experimental details, including model and optimization settings are listed in Table 2.

# E    Full Visualization of Seizure Evolution Process

To additional illustrate the effectiveness of our proposed *DMNet*, we present the visualization results of the difference matrix throughout the entire seizure process in Fig. 8. The upper figure shows the

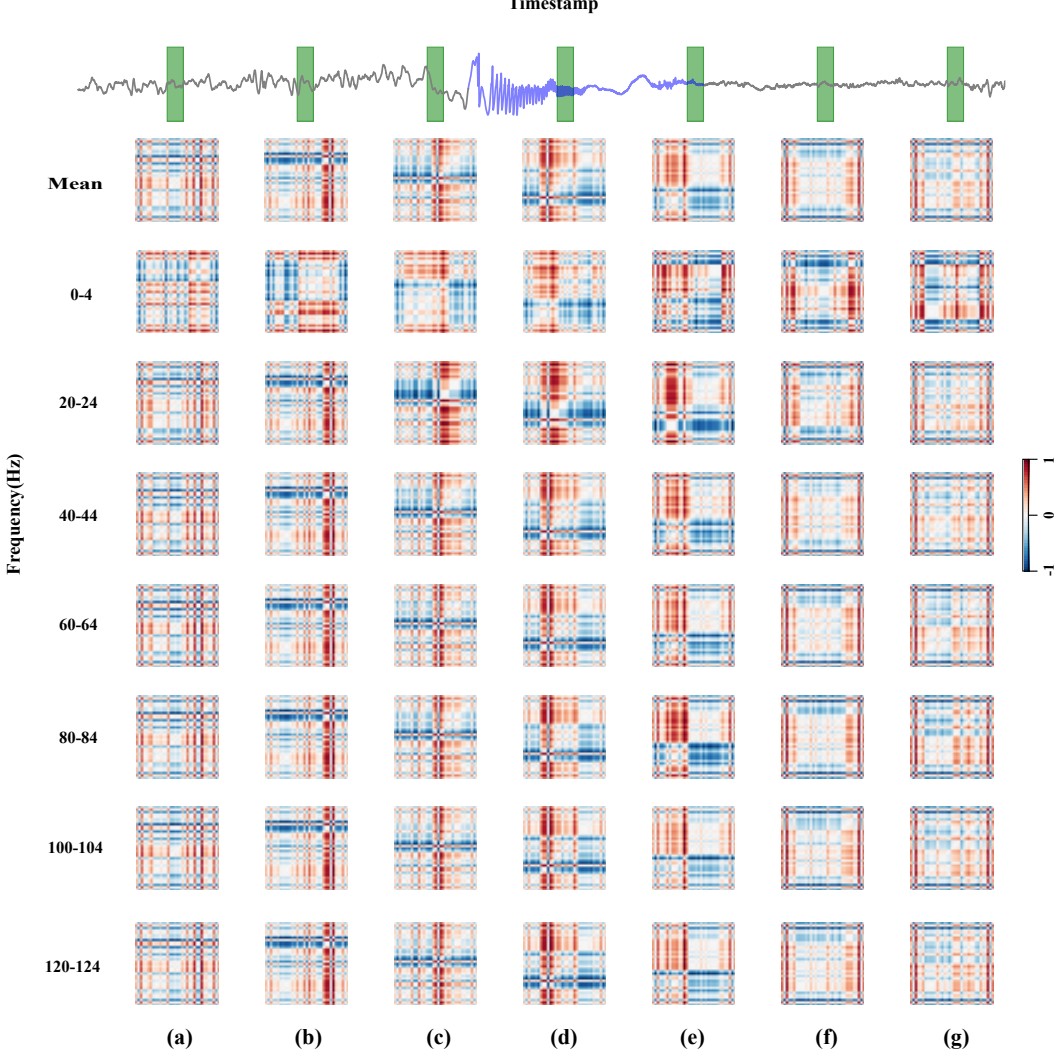

Figure 8: Illustration of how difference matrices reflect seizure activity changes during seizure evolution process. Purple line refers to seizure waves, gray line refers to normal waves.

raw iEEG signal that contains a complete seizure process, where the gray wave represents the normal signal and the purple wave represents the seizure. The green masked blocks indicate the segments for detection. Notably, distinctions between seizure and normal matrices are evident during the evolution of the seizure. For segments that are temporally closer to the seizure events, the difference matrices tend to be rougher (e.g., segments c, d, and e) and vice versa (e.g., segments a, b, f, and g). In summary, this case clearly illustrates how the difference matrix captures the seizure activity changes in different frequency bands, and indicates the effectiveness of our proposed method.

## F Generalization Ability Analysis

To further evaluate the generalization capability of DMNet across a wider range of subjects with greater heterogeneity, we assessed the model using data from the extensive TUSZ EEG dataset [1] which encompasses numerous subjects. After data preprocessing, we retained data from 179 subjects, dividing them into training, validation, and testing sets in a 6:2:2 ratio, with distinct subjects in each split. Please refer to Tab. 6 for the results of the experiment. The results indicate that DMNet

Table 6: Performance of DMNet on TUSZ Dataset.

| Model | TUSZ | | | |
|---|---|---|---|---|
| | Pre. | Rec. | F1 | F2 |
| Abou-Abbas et al. | $25.03_{\pm3.881}$ | $57.20_{\pm3.382}$ | $36.27_{\pm4.528}$ | $43.88_{\pm3.181}$ |
| Zhao et al. | $23.74_{\pm4.502}$ | $49.73_{\pm3.467}$ | $30.28_{\pm4.022}$ | $42.41_{\pm3.285}$ |
| Dissanayake et al. | $28.67_{\pm3.294}$ | $54.91_{\pm4.492}$ | $36.70_{\pm2.587}$ | $49.15_{\pm5.128}$ |
| SICR | $32.54_{\pm4.621}$ | $61.32_{\pm5.062}$ | $40.71_{\pm3.296}$ | $50.45_{\pm4.957}$ |
| SEEGNet | $34.16_{\pm4.381}$ | $58.20_{\pm5.124}$ | $45.34_{\pm4.213}$ | $52.63_{\pm5.648}$ |
| PPi | $36.49_{\pm5.682}$ | $62.72_{\pm4.482}$ | $49.69_{\pm5.107}$ | $56.09_{\pm4.784}$ |
| DMNet | $\mathbf{48.87}_{\pm2.520}$ | $\mathbf{67.59}_{\pm4.834}$ | $\mathbf{56.56}_{\pm3.347}$ | $\mathbf{65.86}_{\pm2.972}$ |

Table 7: Full average performance with standard deviation of of subject-independent seizure detection tasks on clinical dataset. The **v** indicates the first in a column and v indicates the second.

| Model | Clinical | | | |
|---|---|---|---|---|
| | Pre. | Rec. | F1 | F2 |
| SelfReg | $51.60_{\pm3.234}$ | $48.74_{\pm4.185}$ | $\underline{51.24}_{\pm3.138}$ | $48.63_{\pm2.980}$ |
| GroupDRO | $47.60_{\pm1.760}$ | $44.74_{\pm3.123}$ | $45.15_{\pm1.920}$ | $46.33_{\pm1.990}$ |
| MTL | $20.46_{\pm3.350}$ | $52.33_{\pm4.040}$ | $28.59_{\pm3.128}$ | $39.13_{\pm2.963}$ |
| CORAL | $38.70_{\pm2.455}$ | $49.20_{\pm3.893}$ | $42.01_{\pm2.620}$ | $47.66_{\pm2.680}$ |
| CDANN | $33.43_{\pm5.783}$ | $40.41_{\pm5.050}$ | $35.72_{\pm5.013}$ | $37.58_{\pm4.668}$ |
| SD | $18.69_{\pm2.475}$ | $\underline{54.40}_{\pm3.823}$ | $28.78_{\pm2.600}$ | $40.81_{\pm2.645}$ |
| IB_IRM | $29.19_{\pm1.948}$ | $49.75_{\pm3.785}$ | $37.91_{\pm2.035}$ | $42.64_{\pm2.208}$ |
| VREx | $44.80_{\pm2.213}$ | $32.45_{\pm3.745}$ | $36.33_{\pm2.450}$ | $35.34_{\pm2.613}$ |
| IB_ERM | $40.30_{\pm1.668}$ | $37.40_{\pm3.338}$ | $37.59_{\pm1.880}$ | $37.19_{\pm2.125}$ |
| TRM | $34.03_{\pm2.023}$ | $42.74_{\pm4.285}$ | $38.93_{\pm2.448}$ | $41.58_{\pm3.003}$ |
| Abou-Abbas et al. | $43.24_{\pm5.313}$ | $45.84_{\pm4.433}$ | $43.15_{\pm4.718}$ | $46.95_{\pm4.550}$ |
| Zhao et al. | $30.17_{\pm3.630}$ | $49.44_{\pm5.893}$ | $36.16_{\pm4.168}$ | $42.65_{\pm4.943}$ |
| Dissanayake et al. | $40.12_{\pm4.853}$ | $39.29_{\pm6.493}$ | $38.30_{\pm5.783}$ | $40.82_{\pm6.160}$ |
| SICR | $46.27_{\pm5.655}$ | $43.91_{\pm6.253}$ | $45.65_{\pm5.380}$ | $43.86_{\pm5.695}$ |
| SEEGNet | $44.89_{\pm7.073}$ | $47.70_{\pm4.355}$ | $46.25_{\pm6.432}$ | $47.11_{\pm4.570}$ |
| PPi | $\underline{51.72}_{\pm6.847}$ | $49.70_{\pm5.712}$ | $49.78_{\pm4.050}$ | $51.12_{\pm4.293}$ |
| DMNet | $\mathbf{59.58}_{\pm4.648}$ | $\mathbf{55.24}_{\pm5.120}$ | $\mathbf{54.49}_{\pm3.915}$ | $\mathbf{55.93}_{\pm3.751}$ |
| DMNet w/o $L_{c\ell}$ | $48.25_{\pm4.110}$ | $53.30_{\pm3.859}$ | $49.62_{\pm3.858}$ | $\underline{51.20}_{\pm3.096}$ |
| DMNet w/o $L_c$ | $51.39_{\pm3.820}$ | $47.43_{\pm4.107}$ | $47.32_{\pm4.953}$ | $47.15_{\pm4.120}$ |
| DMNet w/o **DM** | $43.58_{\pm4.562}$ | $45.79_{\pm3.870}$ | $46.72_{\pm3.205}$ | $43.42_{\pm4.823}$ |

consistently outperforms existing SOTA models, demonstrating its effectiveness in seizure detection on EEG dataset with numerous subjects.

# G Full Result

# H Limitations

There are two main limitations to my approach. Firstly, it lacks a theoretical foundation, which may call into question the effectiveness and reliability of the method. It becomes challenging to explain and justify the underlying principles behind the approach. Secondly, although the method is efficient, it suffers from a limited number of parameters. This limitation can potentially impact the generalizability of the method. Therefore, while the method may show promising results in specific contexts, caution should be exercised when applying it to broader or unfamiliar situations.

Table 8: Full average performance with standard deviation of of subject-independent seizure detection tasks on MAYO. The **v** indicates the first in a column and v indicates the second.

| Dataset / Model | MAYO | | | |
|---|---|---|---|---|
| | Pre. | Rec. | F1 | F2 |
| SelfReg | $60.40_{\pm 5.788}$ | $32.13_{\pm 4.320}$ | $36.13_{\pm 4.423}$ | $32.12_{\pm 3.988}$ |
| GroupDRO | $48.31_{\pm 7.228}$ | $35.00_{\pm 9.053}$ | $27.82_{\pm 6.870}$ | $28.04_{\pm 6.823}$ |
| MTL | $46.87_{\pm 6.840}$ | $22.08_{\pm 7.813}$ | $15.68_{\pm 4.915}$ | $16.31_{\pm 5.003}$ |
| CORAL | $62.17_{\pm 7.843}$ | $29.86_{\pm 9.835}$ | $20.41_{\pm 7.335}$ | $22.01_{\pm 7.430}$ |
| CDANN | $36.79_{\pm 5.453}$ | $79.55_{\pm 4.335}$ | $45.49_{\pm 5.578}$ | $56.60_{\pm 4.843}$ |
| SD | $47.73_{\pm 6.518}$ | $55.59_{\pm 4.735}$ | $46.97_{\pm 4.825}$ | $50.35_{\pm 4.485}$ |
| IB_IRM | $47.57_{\pm 7.173}$ | $57.17_{\pm 5.315}$ | $46.86_{\pm 6.093}$ | $50.71_{\pm 5.490}$ |
| VREx | $51.21_{\pm 6.155}$ | $59.95_{\pm 5.323}$ | $51.19_{\pm 5.738}$ | $54.85_{\pm 5.503}$ |
| IB_ERM | $46.29_{\pm 6.553}$ | $57.36_{\pm 4.405}$ | $47.21_{\pm 5.303}$ | $51.44_{\pm 4.725}$ |
| TRM | $47.55_{\pm 7.498}$ | $58.97_{\pm 4.785}$ | $43.96_{\pm 4.788}$ | $47.87_{\pm 3.473}$ |
| Abou-Abbas et al. | $48.47_{\pm 4.898}$ | $51.40_{\pm 5.690}$ | $50.69_{\pm 4.985}$ | $48.86_{\pm 5.290}$ |
| Zhao et al. | $37.07_{\pm 4.344}$ | $56.06_{\pm 4.686}$ | $26.17_{\pm 4.367}$ | $38.48_{\pm 4.870}$ |
| Dissanayake et al. | $50.39_{\pm 6.033}$ | $68.99_{\pm 4.323}$ | $57.69_{\pm 3.838}$ | $64.82_{\pm 3.173}$ |
| SICR | $\mathbf{79.01}_{\pm 3.980}$ | $63.29_{\pm 3.058}$ | $69.88_{\pm 2.208}$ | $66.17_{\pm 2.623}$ |
| SEEGNet | $71.82_{\pm 7.341}$ | $60.50_{\pm 3.091}$ | $64.87_{\pm 3.421}$ | $63.15_{\pm 3.340}$ |
| PPi | $\underline{74.49}_{\pm 8.550}$ | $70.21_{\pm 2.725}$ | $\underline{72.28}_{\pm 3.915}$ | $71.02_{\pm 3.070}$ |
| DMNet | $68.82_{\pm 7.168}$ | $\mathbf{90.06}_{\pm 1.270}$ | $\mathbf{73.08}_{\pm 3.738}$ | $\mathbf{81.54}_{\pm 3.283}$ |
| DMNet w/o $\boldsymbol{L_{c\ell}}$ | $47.10_{\pm 4.055}$ | $\underline{89.15}_{\pm 3.688}$ | $62.63_{\pm 2.243}$ | $\underline{76.43}_{\pm 5.543}$ |
| DMNet w/o $\boldsymbol{L_c}$ | $58.34_{\pm 4.825}$ | $76.73_{\pm 4.608}$ | $64.28_{\pm 4.553}$ | $71.79_{\pm 4.823}$ |
| DMNet w/o **DM** | $49.67_{\pm 5.303}$ | $71.54_{\pm 4.553}$ | $60.63_{\pm 4.825}$ | $66.57_{\pm 2.808}$ |

Table 9: Average performance with standard deviation of of subject-independent seizure detection tasks on FNUSA. The **v** indicates the first in a column and v indicates the second.

| Dataset / Model | FNUSA | | | |
|---|---|---|---|---|
| | Pre. | Rec. | F1 | F2 |
| SelfReg | $62.54_{\pm 4.373}$ | $48.19_{\pm 6.663}$ | $49.20_{\pm 4.413}$ | $47.73_{\pm 5.630}$ |
| GroupDRO | $53.48_{\pm 4.318}$ | $71.44_{\pm 3.943}$ | $60.47_{\pm 3.728}$ | $66.38_{\pm 3.665}$ |
| MTL | $60.04_{\pm 4.088}$ | $52.64_{\pm 5.173}$ | $53.90_{\pm 4.298}$ | $52.83_{\pm 4.780}$ |
| CORAL | $\mathbf{65.13}_{\pm 3.615}$ | $53.23_{\pm 5.395}$ | $55.93_{\pm 4.283}$ | $53.88_{\pm 4.930}$ |
| CDANN | $\underline{64.37}_{\pm 5.103}$ | $54.85_{\pm 5.558}$ | $54.35_{\pm 4.608}$ | $53.86_{\pm 4.520}$ |
| SD | $56.99_{\pm 4.583}$ | $57.97_{\pm 4.970}$ | $55.42_{\pm 3.688}$ | $56.45_{\pm 4.240}$ |
| IB_IRM | $54.22_{\pm 4.985}$ | $63.26_{\pm 4.608}$ | $55.96_{\pm 3.503}$ | $59.47_{\pm 3.768}$ |
| VREx | $54.74_{\pm 4.923}$ | $60.15_{\pm 5.458}$ | $54.64_{\pm 4.010}$ | $57.12_{\pm 4.553}$ |
| IB_ERM | $54.64_{\pm 4.948}$ | $55.26_{\pm 5.363}$ | $52.68_{\pm 4.178}$ | $53.70_{\pm 4.690}$ |
| TRM | $60.68_{\pm 4.240}$ | $58.46_{\pm 6.150}$ | $56.00_{\pm 4.325}$ | $56.74_{\pm 5.213}$ |
| Abou-Abbas et al. | $49.83_{\pm 4.215}$ | $56.90_{\pm 4.368}$ | $52.33_{\pm 3.923}$ | $57.53_{\pm 4.123}$ |
| Zhao et al. | $41.64_{\pm 3.050}$ | $44.20_{\pm 2.168}$ | $40.62_{\pm 0.798}$ | $42.12_{\pm 1.123}$ |
| Dissanayake et al. | $63.85_{\pm 6.980}$ | $76.01_{\pm 3.873}$ | $63.75_{\pm 4.368}$ | $67.94_{\pm 2.598}$ |
| SICR | $63.78_{\pm 3.238}$ | $66.77_{\pm 4.078}$ | $64.25_{\pm 3.363}$ | $65.10_{\pm 3.748}$ |
| SEEGNet | $62.23_{\pm 4.070}$ | $72.35_{\pm 2.620}$ | $\underline{66.81}_{\pm 4.337}$ | $68.92_{\pm 3.832}$ |
| PPi | $59.53_{\pm 4.823}$ | $75.42_{\pm 3.021}$ | $65.83_{\pm 4.981}$ | $\underline{71.59}_{\pm 4.515}$ |
| DMNet | $62.30_{\pm 5.475}$ | $\mathbf{85.39}_{\pm 2.623}$ | $\mathbf{67.80}_{\pm 3.795}$ | $\mathbf{75.15}_{\pm 2.460}$ |
| DMNet w/o $\boldsymbol{L_{c\ell}}$ | $52.57_{\pm 6.623}$ | $\underline{78.49}_{\pm 5.323}$ | $60.99_{\pm 3.995}$ | $70.75_{\pm 3.908}$ |
| DMNet w/o $\boldsymbol{L_c}$ | $49.28_{\pm 6.155}$ | $73.48_{\pm 4.418}$ | $58.12_{\pm 3.723}$ | $65.23_{\pm 4.686}$ |
| DMNet w/o **DM** | $46.98_{\pm 6.225}$ | $66.89_{\pm 4.206}$ | $56.21_{\pm 4.195}$ | $62.18_{\pm 4.846}$ |

