# OpenReview forum: "DMNet: Self-comparison Driven Model for Subject-independent Seizure Detection"
_NeurIPS.cc/2024/Conference — NeurIPS 2024 poster_

### Official Review · Reviewer_NPyS · 2024-07-12

**Soundness:** 4
**Presentation:** 4
**Contribution:** 4
**Rating:** 9
**Confidence:** 4

**Summary:**

The authors present an algorithms for subject-independent automatic seizure detection. the algorithm exploits the dynamic behavior of seizures by including a contextual region of analysis, and a channel reference, where contextual refer to the time window close to the window of analysis, and the channel reference to the long term average behavior of the signal. They propose to capture the differences in this dynamic behavior, in the frequency domain, in the Difference Matrix, which is then input in a CNN for its classification. The authors show the results for three different datasets, 2 public and one private, compared with different state of the art algorithms for seizure detection. They demonstrate that DMNet outperforms other algorithms.

**Strengths:**

1. The propose a novel algorithm for independent-subject automatic seizure detection.
2. They consider the use of dynamic information, and changes in this dynamic, in order to mitigate the effect of inter-subject variability. This is clever since the nature of the seizures is dynamic.
3. They validate the algorithm in different datasets, and using different SOTA algorithms.
4.

**Weaknesses:**

1. Can you indicate the sampling frequency of the signals analyzed? Also, you make the analysis based on number of samples, but I do believe this is dependent on the sampling frequency. For instance, if I evaluate the length of segment L for a signals sampled at 250Hz, or a signal sampled at 1KHz, the results with respect to performance metrics might be different. Can you elaborate a bit more about this? Will your algorithm work also for different sampling frequencies, or it should be fine tuned?


minor:

1. Can you improve the legibility of the figures, perhaps using a larger font in some cases and/or using alight font instead of a bold one in other cases.
2. Line 85 use named or called, since namely is used to express the specific details of something but not to assign a name.
3. I considere that the deployment of an online system should be listed as a separate contribution, since merging the third contribution with this mitigates negatively the impact of this item.
4. Line 162, I think you refer to figure 2c and 2d, and not to figure 3.

**Questions:**

1. The channel-level reference also contain information of seizures, would it be possible to generate this reference pattern only for free of seizures segments? What is the possible impact of having information in this reference free of seizures, or containing seizures? Can you elaborate about this?
2. Is a linear classifier good enough to discriminate between the latent representation Z, for normal and seizure segments? Did you also explore the use of other classifiers?.

---

> ### Author Rebuttal · Authors · 2024-08-07
>
> **w1: Discussion of the impact of different frequencies of data on the analysis and model performance.**
>
> We thank the reviewer for this thoughtful question. The public dataset has the original sampling frequency of 5000Hz; then we downsample the data to 2500Hz. For the clinical dataset, the original sampling frequency varies from 512Hz to 1024Hz for different subjects (detailed descriptions in Table 3 in Appendix D); then we downsample the data to 250Hz (please refer to Appendix E, line 630).
>
> For segments of the same length ($L$) but with different sampling frequencies (e.g., segment A with 250Hz **vs** segment B with 1000Hz), the frequency domain representation vector of segment A would have a smaller size than that of segment B due to the lower frequency resolution. This means that segment B can preserve richer neural activity information within the high-frequency range.
>
> On the contrary, due to the same length of L, segment B would cover a shorter time duration than segment A, leading to limitations in capturing the long dependencies between seizure events and their contexts. To address this issue, a simple way is to increase the length of segment ($L$) for segment B so that it can cover a longer time duration.
>
> If the model is directly applied to frequencies different from those in its training dataset, fine-tuning on the new frequency data is required. Another approach is to align the frequencies of the dataset for inference by upsampling or downsampling it to match the frequency of the training set, in which case fine-tuning is not necessary. However, there may be a performance penalty.
>
> In our work, although the richer information in the high-frequency domain would yield better performance, it reduces algorithm efficiency. We argue that, compared to information within high-frequency bands, the duration that the reference segments cover is more important, as the long dependencies between seizure events and their contexts are essential to model the semantic information about the evolution of seizures. Therefore, as a trade-off between performance and efficiency, we uniformly downsample the original data.
>
> **w2: Minors**
>
> We really appreciate the review for the careful reading and suggestions, and we'll make the modifications in our final version including:
>
> \- Uniformly using a larger and alight font in cases.
>
> \- Replacing "namely" with "named"  in Line 85.
>
> \- Listing the deployment of an online system as a separate contribution.
>
> \- Correcting the figure 2c and 2d referings in Section 3.2.
>
> **q1: Discussion on the effect of channel-reference including or not including seizures on model performance.**
>
> We thank the reviewer for this insightful question. In our dataset, the ratio of seizure segments is very low, making the impact of included seizure segments during clustering for channel-level reference generation negligible. However, we agree that generating this reference pattern exclusively with normal segments would likely improve performance. Incorporating more seizure segments in the channel-level reference could result in a less distinguishable seizure pattern.
>
> **q2. Discussion about the classifier in DMNet.**
>
> We thank the reviewer for this insightful question. Unfortunately, the linear classifier may not suffice to discriminate the seizure events since the generated representation of difference matrix would contain the non-lieanr patterns (Please refer to Figure 8 in Appendix for a more detailed presentation).

---

> > ### Comment · Reviewer_NPyS · 2024-08-13
> >
> > I thank the authors for their reply to my inquiries. I am satisfied with the answers and have no further comments.

---

> > > ### Author Response · Authors · 2024-08-13
> > > **Thank you**
> > >
> > > We truly appreciate your effort in helping us to strengthen the paper and your support for our work.

---

### Official Review · Reviewer_EpQU · 2024-07-12

**Soundness:** 3
**Presentation:** 3
**Contribution:** 2
**Rating:** 5
**Confidence:** 3

**Summary:**

The paper presents a seizure detection approach for spotting seizure segments in long recordings by comparing the differences in spectral content between the target segment to be labeled and surrounding context segments and some prototypical context segments, obtained as centroids by clustering the channel, along the differences between these segments organized into a matrix. A CNN based network then processes this matrix using the distribution of differences to understand if the differences are indicative of a seizure.  Results show state-of-the-art performance a multiple datasets.

**Strengths:**

The paper is easy to read with helpful figures and only minor confusions.

The paper appears to achieve the state of the art performance in seizure detection.

**Weaknesses:**

There are few points that are not clear  (see questions).

My main concern is that when looking at the appendix, the data division and hyper-parameter selection is not clear. There is a set of basic hyper-parameters that in Table 2 that seems to be set globally, but now clear how. Then there is a validation set in each split. Perhaps this validation split is used just for choosing the stoping criterion. To choose the best hyper-parameters based on test set performance across all folds is not a valid approach (as it uses test set performance to choose a model architecture). A fair approach would be conduct internally the hyper-parameter selection based on validation in each fold.

Overall, there is lack of intuition (or theory) on when and why organizing the total set of differences into a matrix makes sense. Although it is meaningful to organize neighboring segments there isn't any spatial ordering between centroids differences.  It would seem that multiple branches to compare the none contiguous with 1D convolutions (across frequencies and then channels) before  CNN (bu

**Questions:**

Line 82: "of normal and " -> "of normal segments and"

In Figure 2(b) it looks like the data has not be high-pass filtered based on the cross-time analysis.  How would high pass filtering affect the proposed methodology as well as the challenges in existing data?

Line 160, as the discrete Fourier transform will be complex valued, it is the sum of the magnitude of the differences, unless only the amplitudes are preserved. Notably, by Parseval's theorem the sum of the squared magnitudes of the differences is the same as the sum of the squared magnitude in the time domain. Thus, as the phase may vary it should be the absolute value of the differences in the magnitude (or squared magnitude).  Later on 196 these are stated to be $d$-dimensional real vectors so the reader can finally assume they are magnitudes (which are non-negative).

I don't understand lines 214–217, why would the clusters have left and right meaning and why would duplicating them in reverse order matter in the construction of the difference matrix?

While the left and right windows have ordered information its still not clear why appending the cluster centroids to both sides helps or why a single vector representation of the difference of the target to the rest couldn't be used. The reader is left to guess that the additional differences simply provide context for the differences to the target.

Figure 3 is not sufficient for the neural network architecture. How does the "global average pooling"  operate as it seems to take the two previous maxPool2D layers along with the output of the 2 by 2 max pooling... How was this design chosen?

 As mentioned in the limitations in the appendix, the method may be limited due to limited parameters to tune during training. I have two suggestions:
1. Use an ablation on the 2D matrix vectorizing the redundant differences and using a network with full connected layers rather than the matrix organized version processed with a CNN.
2. Learn weightings on the frequencies.

**Limitations:**

Limited by unclear operation.

---

> ### Author Rebuttal · Authors · 2024-08-07
>
> **w1: Clarify the data division and hyper-parameter selection.**
>
> We apologize for overlooking the detailed for the data division and hyper-parameter selection. Detailed settings please refer to **GR2** in **Global Response.** We will update this part in the manuscript.
>
> **w2: Discuss the intuition (or theory) on when and why organizing the total set of differences into a matrix makes sense.**
>
> Thank you for the insightful question. The detail explanation please refer to **GR3** in **Global Response.**
>
> **q1: Discuss how high-pass filtering affects the proposed methodology and the challenges in existing data.**
>
> Thank you for raising this question. Before analyzing most EEG signals, it is necessary to filter the brain's high-frequency signals. However, this preprocess will confuse the model in seizure detection. For example，Some other brain signal components, such as spikes, artifacts or other sharp activities that are not band limited and whose energy can expand to high frequencies may disturb the detection process by generating false oscillations in the filtered signal. That is because by passing through a filter, sharp transients result in short-duration oscillations (close to the filter impulse response) that may easily be confused with real oscillations.  Detailed research conclusions can be found in [1].
>
> **q2: Clarify whether complex numbers or magnitudes are used after the Fourier transform.**
>
> Thank you so much for pointing that out. Your analysis is very correct, and we are very sorry that we did not make it clear that we use magnitudes after Fourier transforms are applied to sequence segments. We will update this statement in the final version of the manuscript.
>
> **q3: Clarify why reverse the L_cl to form R_cl and concatenate it on the right.**
>
> Thanks for pointing this out. The detail explanation please refer to **GR1** in **Global Response.** We will update this expression in the manuscript.
>
> **q4: Clarify the advantages of using 2-d difference matrix**
>
> Thank you for raising this question. As you mentioned, the additional differences provide a context for a richer representation.
>
> We construct a 2D difference matrix to facilitate the neural network in capturing two types of information: 1. the difference between the target segment and other segments, and 2. the differences between the differences calculated from different segments. A standalone 1D vector only contains the difference information between the target and other segments, lacking local (contextual reference) or global (channel-level reference) differences with other positions.
>
> **q5: Clarify how the 'global average pooling' operates on the input and why.**
>
> Thanks for pointing out that Figure 3 may lack some details to fully grasp the architecture, especially with regard to global average pool operations. Let's break down the confusion and address the design choice. We will update the Figure 3 and related expression in the manuscript.
>
> 1. **Clarifying Global Average Pooling**
>
> The statement that global average pooling "takes the two previous maxPool2D layers along with the output of the 2 by 2 max pooling" is inaccurate. Here's how global average pooling actually works within this architecture:
>
> Independent Operation: Global average pooling operates independently on each of the three convolutional pathways. It doesn't combine outputs from different stages.
>
> Input to Global Average Pooling: The input to the global average pooling layer in each pathway is solely the output of the last convolutional/pooling operation within that specific pathway. For instance:
>
> Pathway 1: Input to global average pooling is the output from the 1-th MaxPool2D.
>
> Pathway 2: Input to global average pooling is the output from the 2-th MaxPool2D.
>
> Pathway 3: Input to global average pooling is the output from the 3-th Conv2D.
>
> Mechanism: Global average pooling calculates the average value of each feature map independently. If a feature map at the input of global average pooling has dimensions H x W, it is reduced to a single scalar value representing the average across all H x W positions.
>
> 2. **Design Choice Rationale**
>
> The choice of using global average pooling in this manner is likely motivated by the following factors:
>
> **Integration of Multi-Scale Features:** Each convolutional pathway processes different receptive field sizes to extract features at various scales. For instance, paths that undergo multiple pooling layers can capture more global features, while paths with fewer pooling layers retain more local details.
>
> Integrating the Global Average Pooling (GAP) results from each pathway is akin to merging information from different scales, resulting in a more comprehensive and enriched feature representation. This process facilitates the model in making more precise judgments.
>
> **Feature Summarization:** Global average pooling acts as a way to summarize the information present in each feature map into a single representative value. This is particularly useful for converting convolutional feature maps into a fixed-length vector, which is necessary for feeding into fully connected layers or for tasks like classification.
>
> **Regularization:** By reducing the number of parameters compared to using fully connected layers for this summarization, global average pooling can help prevent overfitting.
>
> **q6. Suggestions on how to improve the model.**
>
> We thank the reviewer for this insightful suggestions. we agree that the linear layer also makes sense and will add this discussion to the limitation of the paper. Moreover, learning weightings on the frequencies is a promising research idea, considering that seizure patterns differ from normal patterns in the frequency domain. I will further explore this idea in the future.
>
> [1] C.G. Bénar, L. Chauvière, F. Bartolomei, F. Wendling,Pitfalls of high-pass filtering for detecting epileptic oscillations: A technical note on “false” ripples, Clinical Neurophysiology

---

> > ### Comment · Reviewer_EpQU · 2024-08-11
> >
> > Thank you for the response. I think some answers are satisfactory except those in the global response, where I ask more.
> >
> > I think the paper should acknowledge that the twice concatenated channel-wise references in the  2D matrix of differences is an heuristic choice that tries to put a "round peg in a square hole". As in reality, the local operations in the CNN only make sense for the contextual references. A Transformer architecture that processes differences seems to make more sense.
> >
> > I'll consider raising my score once the hyper-parameter selection process is clarified.

---

> ### Author Response · Authors · 2024-08-12
> **Settings of Cross-dataset Experiment in rebuttal**
>
> We thank the reviewer for all the insightful comments. I will revise the paper to acknowledge that the decision to concatenate channel-wise references twice in the 2D matrix of differences is a heuristic choice.
>
> Moreover, responses to specific comments are listed below.
>
> ### Settings of Cross-dataset Experiment in rebuttal.
>
> Regarding the additional cross-dataset experiments, for the source dataset, we first randomly devide the subjects into several groups. We take the case of Clinical dataset (it will be divided into 4 groups) as source domain as an example for detailed illustration.
>
> To minimize experimental variability, we assign each group as a validation set (1 group) and the rest as a training set (comprising 3 groups), resulting in 4 experiments (listed below).
>
> Table-1:
>
> | Experiment   | Training Set (3 groups)   | Validation Set (1 group) |
> |:------------:|:-------------------------:|:------------------------:|
> | Experiment 1 | Group 1, Group 2, Group 3 | Group 4                  |
> | Experiment 2 | Group 1, Group 2, Group 4 | Group 3                  |
> | Experiment 3 | Group 1, Group 3, Group 4 | Group 2                  |
> | Experiment 4 | Group 2, Group 3, Group 4 | Group 1                  |
>
> Next, for each experiment, we implement LODO on the training set (comprising 3 groups) to create a sub-training set (2 groups) and a sub-validation set (1 group) for hyperparameter selection. Finally, each experiment has 3 sub-experiments to select the hyperparameters. The 3 sub-experiments of experiment 1 are listed below, and the other experiments are constructed in a similar manner.
>
> Table-2:
>
> | Sub-Experiment   | Sub-Training Set (2 groups) | Sub-Validation Set (1 group) |
> |:----------------:|:---------------------------:|:----------------------------:|
> | Sub-Experiment 1 | Group 1, Group 2            | Group 3                      |
> | Sub-Experiment 2 | Group 1, Group 3            | Group 2                      |
> | Sub-Experiment 3 | Group 2, Group 3            | Group 1                      |
>
> In this case, for each experiment, every set of parameters would yields 3 performance scores on 3 sub-experiments respectively, and then we select the set of hyperparameters achieving the best score as the global parameters for model implementation. So strictly speaking in our process, we regard group (containing 2 subjects) as a domain.
> During model implementation, for each experiment in Table-1, we train the model using the training set of the clinical dataset and employ an early stopping strategy based on the validation set loss. The trained model is then utilized for evaluate the performance on the dateset of other two datasets (MAYO, FNUSA).
> Please note that this process ensures the hyper-parameters are searched independently without any exposure to the test set. The final hyperparameter configurations for the three experiments are as follows:
>
> Table-3:
>
> | Source Domain | Clinical | FNUSA    | MAYO     |
> |:----------------------------------------- |:--------:|:--------:|:--------:|
> | Length of Segment                         | 250      | 450      | 500      |
> | Number of Segment                         | 12       | 8        | 8        |
> | Number of Cluster                         | 8        | 7        | 7        |
> | Base filter Number                        | 64       | 16       | 8        |
> | Learning Rate                             | 3.00E-04 | 3.00E-04 | 3.00E-04 |
> | Batch Size                                | 32       | 24       | 24       |

---

> ### Author Response · Authors · 2024-08-12
> **Hyperparameter Selection in Experiments in Our Main Body**
>
> ### Hyperparameter Selection in Experiments in Our Main Body.
>
> For global hyper-parameter selection, we first randomly divide the dataset into different groups, as detailed in Table 4 of the Appendix. We then conduct trials according to the setup in Table 5 of the Appendix, with each trial involving distinct groups for training, validation, and testing.
>
> Please note that for each subject, their SEEG signal is further divided into two parts based on time, with the first 20% of the data extracted for validation and the remaining data used for testing. However, when the data is used as part of the training set, the entire dataset is utilized.
>
> The entire trial process includes two main procedures:
>
> 1. Global hyper-parameter selection.
>    For global hyper-parameter selection, we train the model and evaluate the performance score on the training and validation group pairs for each trial (e.g., 12 pairs in total for the Clinical dataset). We then select the set of hyper-parameters that achieve the best average scores across all group pairs, thereby determining the global hyper-parameters. The set of hyper-parameters is listed in Table 2 in Appendix.
>
> 2. Model training and testing under the selected global hyper-parameters.
>    In the model training and testing process, for each trial, we deploy DMNet with the selected global hyper-parameters on the training (source) group for model training and evaluate the performance on the validation set. The model that achieves the best scores on the validation set is then used to evaluate performance on another test set.
>
> It's important to note that by dividing the validation and testing segments for each subject, we ensure that the test data remains strictly unseen during global hyper-parameter selection. One potential risk is domain information leakage, as global hyper-parameter selection inevitably allows the learner to have a preliminary view of a small portion of data across all domains. However, we argue that this trade-off is acceptable for the following reasons:
>
> - Global hyper-parameter selection facilitates the practical deployment of deep learning models in real clinical scenarios and enhances domain generalization. Specifically, it enables us to identify a more general set of settings that are potentially suitable for most domains. This allows us to directly deploy the model with these global settings for a new patient without the need for hyper-parameter searching or training the model from scratch, greatly improving the scalability of deep learning models in real applications. Please note that some previous works [1,2], global hyper-parameters are also utilized for cross domain testing.
>
> - Using global hyper-parameters ensures a fairer performance comparison across different domains. If each fold (trial) uses different hyper-parameters, performance discrepancies might arise from the hyper-parameters rather than the model itself.
>
> Additionally, in our added experiments of rebuttal (PDF, Table 1 - Table 4), we conducted the additional cross-dataset experiments. In these experiments, since the distinct datasets for training&validation and testing, we have that the testing data and domain information is strictly independent with global hyper-parameter selection. The results indicates that our model DMNet also presents the superior performance compared to other baselines.
>
> [1] Yuan et al,. PPi: Pretraining Brain Signal Model for Patient-independent Seizure Detection, NeurIPS'23
>
> [2] Caiet al,. MBrain: A Multi-channel Self-Supervised Learning Framework for Brain Signals, KDD'23

---

> > ### Comment · Reviewer_EpQU · 2024-08-12
> >
> > Thank you for the efforts to address my concerns—they are mostly addressed. I agree that global hyper-parameter selection can cause information leakage but avoiding it requires different hyper-parameters per model.  I would like to highlight that this detailed methodology was not provided in the original submission. I find no mention of chronological split between validation and test in the original manuscript pertaining to  "Please note that for each subject, their SEEG signal is further divided into two parts based on time, with the first 20% of the data extracted for validation and the remaining data used for testing." It is not clear to me that all of the baselines follow the same methodology. It would seem unfair to use a more involved approach for hyper-parameter selection when the goal is really to tell if the proposed architecture is significantly better.  Can this be confirmed for all of the baselines?

---

> > > ### Author Response · Authors · 2024-08-12
> > > **Rebuttal by Authors**
> > >
> > > Thank you for your thorough review and for acknowledging our efforts to address your concerns. We appreciate your feedback and would like to address the remaining points you've raised.
> > >
> > > **Chronological Split.** We sincerely apologize for the oversight in not explicitly mentioning the chronological split between validation and test sets in our original manuscript. This was an unintentional omission on our part. We will add this crucial information to our experiment section in our revised manuscript, clearly stating: "For each subject, their iEEG signal is further divided into two parts based on time, with the first 20% of the data extracted for validation and the remaining data used for testing."
> > >
> > > **Consistency Across Baselines.** We appreciate you bringing this important point to our attention. We can confirm that the same methodology, including the chronological split and hyperparameter selection process, was applied consistently across all baselines and our proposed model. This ensures a fair comparison and maintains the integrity of our results. We will add a detailed description in our revised manuscript explicitly stating the consistency of our methodology across all models. This addition ensures transparency and allows for proper replication of our experiments.
> > >
> > > Morevoer, whether a global hyperparameter can be used to achieve good performance across various domains is also a consideration, as this also reflects the model's generalization ability. Therefore, we adopted this setup in our experiments.
> > >
> > > We thank you for your diligence in reviewing our work. Your comments have helped us improve the clarity and rigor of our paper.

---

### Official Review · Reviewer_s7PY · 2024-07-13

**Soundness:** 3
**Presentation:** 4
**Contribution:** 3
**Rating:** 5
**Confidence:** 4

**Summary:**

This paper revolves around subject-independent seizure detection using intracranial electroencephalography (iEEG) signals. The primary challenge is the domain shift in iEEG signals across different subjects, which hinders the generalization of seizure detection models to new subjects. Existing models often fail to adapt to these domain shifts, leading to reduced performance in subject-independent scenarios.

The authors highlight the limitations of existing iEEG models, which struggle with subject-independent seizure detection due to the variability in iEEG signals across individuals. Previous approaches have not effectively addressed the domain shift issue, resulting in suboptimal performance when applied to new subjects. This sets the stage for the need for a novel model like DMNet that can overcome these challenges and improve subject-independent seizure detection.

The authors have employed a self-comparison mechanism within DMNet, allowing the model to compare iEEG signals within the same subject and across different subjects. This mechanism enables DMNet to learn subject-independent representations of seizure patterns, enhancing its generalization capabilities. Additionally, the novel neural network architecture of DMNet is tailored to leverage these self-comparisons efficiently, leading to improved performance in subject-independent seizure detection tasks.

**Strengths:**

1.	DMNet addresses the critical challenge of domain shift across different subjects, a problem that has limited the effectiveness of previous models.
2.	The self-comparison mechanism allows DMNet to learn representations of seizure patterns that are independent of the subject. This feature enhances the model's ability to perform effectively across new subjects without additional training.
3.	The paper presents a novel neural network architecture specifically designed to utilize the self-comparison mechanism efficiently. This architecture is expected to optimize the performance of the seizure detection model, making it more effective in a variety of clinical settings.

**Weaknesses:**

1.	The paper lacks extensive experimental validation across diverse datasets. This limitation restricts the demonstrated generalizability and robustness of DMNet. For stronger validation, experiments on multiple, varied iEEG datasets would be necessary to affirm the model's efficacy across different scenarios.
2.	The paper could benefit from a more thorough explanation of the theoretical foundations of DMNet. Providing detailed assumptions and complete proofs for the methodologies proposed would enhance the rigor of the research and strengthen the credibility of the model’s theoretical underpinnings.
3.	The model's architecture, characterized by a limited number of parameters, may hinder its adaptability and performance in diverse situations. While the simplicity of fewer parameters might benefit specific conditions, it could limit the model’s ability to handle new or complex data effectively. This could affect the model's application in broader or more varied contexts.

**Questions:**

1.	Could you provide more detailed insights into how the domain shift in iEEG signals manifests across different subjects?

2.	How does DMNet specifically address and mitigate this domain shift issue? Are there any theoretical or empirical justifications for the effectiveness of this approach in handling these shifts?

3.	Can you provide a detailed comparison between DMNet and existing iEEG models for subject-independent seizure detection in terms of key performance metrics such as sensitivity, specificity, and computational efficiency?

4.	Can you provide details on the experimental setup, including data preprocessing, model training, hyperparameter tuning, and the evaluation metrics used?

5.	How does DMNet fare in real-world clinical applications for subject-independent seizure detection?

6.	Have there been any pilot studies or practical implementations of DMNet in clinical settings? What were the findings regarding its effectiveness and usability?

7.	What are the potential challenges or limitations in deploying DMNet in real clinical scenarios, and how does the model address these to ensure practical utility and scalability?

**Limitations:**

The authors have adequately addressed the limitations and identified no potential negative societal impacts of their work.

---

> ### Author Rebuttal · Authors · 2024-08-07
>
> **w1: Additional cross-dataset experiments to verify the efficacy of DMNet across different scenarios.**
>
> Thank you for the good suggestion. I conducted 3 cross-dataset experiments on Clinical, MAYO, and FNUSA. We select one dataset for training and validation set (with distinct subjects), and the other two datasets for testing. For detailed results, please refer to **Tables 2, 3, and 4** in the attached **PDF** within the **Global Response.**
>
> The results showed that DMNet outperforms the existing SoTA, especially in F2. In addition, the performance was degraded relative to experiments with cross subject on a single dataset. This may be attributed to varying annotation standards across different medical institutions, as well as individual annotation preferences among different annotators.
>
> **w2&q2: Discussion of the theoretical foundations of DMNet.**
>
> Thank you for the insightful question. The detail explanation please refer to **GR3** in **Global Response.**
>
> **w3: Discussion  DMNet with few parameters.**
>
> Thank you for your valuable comments on our research. My response consists of 2 parts:
>
> 1. In real-world epilepsy diagnosis, numerous lengthy iEEG files require processing. Given equipment constraints, efficient ML methods are crucial. Although DMNet has a limited number of parameters, it has achieved near-optimal performance on different datasets while maintaining shortest processing times.
>
> 2) Although the model has a limited number of parameters, our design offers good flexibility and scalability. By changing the differential matrix encoder (adopting more complex neural network models), we can further improve model performance to address broader application scenarios.
>
> **q1: Discuss the domain shift in iEEG signals manifests across different subjects**
>
> Thank you for the valuable comment. Our response includes two parts:
>
> 1. **Inter- and intra-subject variability.** The inter-subject variability could be attributed to the factors of age, gender, and living habits, which would be related to the brain topographical and electrophysiology[1]. The intra-subject variability would be explained as the changes of psychological and physiological, such as fatigue, relaxation, and concentration[2].
>
> 2. **iEEG recording experiment. iEEG.** For each subject, doctors need to design an individual electrode implantation based on the subject's profile, including the number and positions of each electrode to be implanted. Variations in electrode placement across subjects can lead to differences in signal properties, such as the specific brain regions being monitored, the distance between electrodes, and the orientation of electrodes relative to neural sources[3]. These differences can cause domain shift in iEEG signals across different subjects.
>
> **q3: Clarify the detailed comparison between DMNet and existing iEEG models.**
>
> I am very sorry that I did not mention that recall is sensitivity and precision is specificity. DMNet significantly outperforms existing iEEG models like SICR, SEEGNet, and PPi in terms of F2, sensitivity (recall) and F1. Detail experiment result please refer to Secion 5.2 in the paper. For the Computational Efficiency, this part can be referred to Section 5.6.
>
> **q4: Clarify details on the experimental setup.**
>
> **Data Preprocessing.** Public datasets are preprocessed by removing power line noise and downsampling to 2500Hz. Clinical datasets undergo similar preprocessing, downsampled to 250Hz. We will include these steps in our manuscript for clarity.
>
> **Data Division in Model Training and Hyperparameter Tuning.** The details please refer to **GR2** in **Global Response.**
>
> **Evaluation Metrics.** We use F2-score as primary metric for performance comparison, because missing any seizure event can be costly in clinical diagnosis. And we also consider Precision, recall, F1-score.
>
> **q5: Discuss how DMNet fares in real-world clinical applications**
>
> DMNet has been successfully deployed in an online system for real-time seizure detection. DMNet can process iEEG data of approximately 10 hours in around 45 seconds, demonstrating its efficiency for real application scenario. Please refer to section 5.6 for a detailed description of the online system.
>
> **q6: Discuss the  pilot studies or practical implementations of DMNet?**
>
> Yes, DMNet has undergone pilot testing in a clinical setting. Compared to older models used in hospitals, DMNet can identify epileptic seizures more quickly and accurately. Notably, based on feedback from doctors, it can process iEEG data in real-time, providing detection results to assist doctors rapidly. This reduces the labor costs associated with doctors having to monitor iEEG data for extended periods.
>
> **q7: Discuss  the potential challenges or limitations in deploying DMNet?**
>
> 1. Incremental Training: Challenges in incremental training emerge when incorporating new subjects, potentially leading to overfitting and knowledge forgetting. We try to apply LoRA[4] for post-training on new subjects. By retaining the original parameters and adjusting only a small additional learnable parameters, we mitigate the risks of overfitting and forgetting.
> 2. Limited Parameters: As subject numbers grow, epilepsy detection complexities rise. Limited model parameters can hinder generalization. We can replace the differential matrix encoder with more complex neural networks to increase the model's parameter capacity for dealing with more complex scenarios.
>
> [1] Seghier ML, Price CJ. Interpreting and Utilising Intersubject Variability in Brain Function. Trends Cogn Sci. 2018
>
> [2] Meyer MC, et al. Electrophysiological correlation patterns of resting state networks in single subjects: a combined EEG-fMRI study. Brain Topogr. 2013
>
> [3] Shi H, et al. Utility of intracranial EEG networks depends on re-referencing and connectivity choice[J]. Brain Communications, 2024.
>
> [4] Hu E , et al. Lora: Low-rank adaptation of large language models[J]. arXiv:2106.09685, 2021.

---

### Official Review · Reviewer_sFSG · 2024-07-15

**Soundness:** 3
**Presentation:** 3
**Contribution:** 2
**Rating:** 5
**Confidence:** 5

**Summary:**

The paper proposes DMNet, a Difference Matrix-based Neural Network for subject-independent seizure detection using intracranial electroencephalography (iEEG). The model addresses the domain shift in iEEG signals across different subjects by leveraging a self-comparison mechanism that aligns iEEG signals and encodes universal seizure patterns. DMNet utilizes contextual and channel-level references to mitigate shifts and employs a difference matrix to capture seizure activity changes effectively. The authors report that DMNet outperforms state-of-the-art models and is highly efficient in real-world clinical applications.

**Strengths:**

1. The proposed self-comparison mechanism seems like a reasonable approach to alleviate inter-subject and intra-subject distribution shift.

2. Presentation is clear and easy to follow, especially in the methodology section, where figures are helpful for understanding. Overall, the paper is well written.

3. Although this model is tailored for iEEG data, it seems straightforward to be generalized for EEG data.

**Weaknesses:**

1. If the right side R_cl merely reverses the left side L_cl, is it redundant?

2. The three datasets used for evaluation have small numbers of subjects. Could the authors include an experiment on EEG data with more subjects?

3. On MAYO and FNUSA, where positive sample ratio is non-trivial, the model has a much higher recall than precision, showing it’s prone to false positives. Do the authors have some preliminary justifications/explanations for this feature?

4. The performance improvement is incremental. In Table 1, if we check F1 score which is a combination of Recall and Precision, we find that on MAYO and FUNSA, the proposed model only outperforms the best baseline for around 1%. Without std reported, we cannot judge is the increment is caused by randomness. On Clinical dataset, the margin is 3% which is good.

**Questions:**

See weakness above.

**Limitations:**

I didn't find a discussion on the limitations of this work.

---

> ### Author Rebuttal · Authors · 2024-08-07
>
> **w1: Clarify why reverse the L_cl to form R_cl and concatenate it on the right**.
>
> Thanks for pointing this out. The detail explanation please refer to **GR1** in **Global Response.**
>
> **w2: Discuss the number of subjects in iEEG dataset  & Additional EEG datasets containing a large number of subjects were added for the experiment.**
>
> 1. Thank you for pointing that out. Obtaining intracranial EEG (iEEG) recordings is challenging as it requires craniotomy surgery for electrode implantation, which involves extensive protocols and approvals. Therefore, in the field of intracranial neural signals, the number of subjects in datasets has not reached the scale of EEG datasets. Some other works in this field also contain a limited number of subjects (e.g., [1] contains 10 subjects, [2] contains 10 subjects). We are also collaborating with medical institutions to release more data, aiming to improve the situation of insufficient data in the field of iEEG.
>
> 2. Thank you for this good suggestion. I have conducted experiments on a large EEG dataset TUSZ[3] with numerous subjects, and after data preprocessing, we retained data from 179 subjects, dividing them into training, validation, and testing sets in a 6:2:2 ratio, with distinct subjects in each split. Please refer to **Table 1** in the attached **PDF** of **Global Response** for the results of the experiment. The results indicate that DMNet consistently outperforms existing SOTA models, demonstrating its effectiveness in seizure detection on EEG dataset with numerous subjects.
>
> **w3: Discuss why the model has a higher recall than precision on 2 public datasets.**
>
> Thank you for this insightful question. The reason our model has a high false positive rate is that the design of DMNet revolves around learning more general seizure patterns. We aim for DMNet to be sensitive to potential seizure events, so that the model minimizes missing any seizures in real clinical applications.  Please note that in most clinical cases, we are more concerned with false negative rates, as the cost of missing positive cases will be far higher than that of false positive cases (for more details, please refer to our response regarding w4). In our results, the higher F2 score of our model also demonstrates the effectiveness of DMNet in identifying seizure events.
>
> In the MAYO and FNUSA datasets,the positive-to-negative sample ratios are 0.21 and 0.36, respectively.  The normal events in MAYO and FNUSA have been down-sampled [4], which will lead to insufficient learning of the general pattern of normal samples, and the model is easy to misjudge normal samples as positive examples, which increases the false positive rate. On the contrary, the clinical dataset was not sampled at all, and the positive and negative sample rate was 0.003. With a large number of normal samples for training, the model could better learn the general representation of normal samples, which made the model have a small false positive rate on the clinical dataset.
>
> **w4: Discuss the performance improvement of DMNet.**
>
> Thank you for pointing that out. Although the improvement of F1 is not significant, the improvement of F2 significantly outperforms all baselines.  Please note that in the field of epilepsy, the F2 metric is more widely employed to evaluate models [1,2]. In addition, the results of std can be found in the appendix.
>
> There are two reasons for why F2 matter in clinical scenario:
>
> 1. In most of clinical scenario, we oftern focus more on the identification of postive event, i.e., emphasis on recall (sensitivity). It is because that the cost of missing a positive case (false negative) would be much higher than the cost of a false positive.
>
> 2. Medical datasets often suffer from significant class imbalance, with the positive class being much rarer than the negative class.
>
> [1]Chen J, Yang Y, Yu T, et al. Brainnet: Epileptic wave detection from seeg with hierarchical graph diffusion learning, KDD2022.
>
> [2]Wang C, *et al.*, "BrainBERT: Self-supervised representation learning for intracranial recordings." ICLR2023
>
> [3]Shah, Vinit et al. “The Temple University Hospital Seizure Detection Corpus.” *Frontiers in Neuroinformatics* 12 (2018): n. pag.
>
> [4]Petr Nejedly, el. Multicenter intracranial EEG dataset for classification of graphoelements and artifactual signals. Scientific Data

---

> > ### Comment · Reviewer_sFSG · 2024-08-12
> >
> > Thanks for the response and I appreciate the new experiments on TUSZ along with the explanations on F2. I'd like to increase my score from 4 to 5.

---

> > > ### Author Response · Authors · 2024-08-12
> > > **Thanks for the reviews**
> > >
> > > We are truly grateful for the reviewer’s feedback and recognition of our efforts.

---

### Author Rebuttal · Authors · 2024-08-07

# Global Response

**GR1. Clarify why reverse the L_cl to form R_cl and concatenate it on the right.**

The additional L_cl (reversed segments of R_cl) enables some originally different seizure sequences to generate similar difference matrix after fully differencing operation, which further enhance the generalization of seizure patterns. A example is provided below to illustrate this point:

Let XY denotes the left channel-reference (L_cl) and YX be the reversed segments as the right. Now  we  consider  two sequences following: 00111 and  11100,  where  0 and 1 represent normal and seizure  events.  Then we  concatenate  the  channel-reference  to  these two  sequences  and  performing  differencing, the results  are  shown  below:

XY00111:
$$ \left[  \begin{matrix}
0&X-Y&X&X&X-1&X-1&X-1\\\\
Y-X&0&Y&Y&Y-1&Y-1&Y-1\\\\
-X&-Y&0&0&-1&-1&-1\\\\
-X&-Y&0&0&-1&-1&-1\\\\
1-X&1-Y&-1&-1&0&0&0\\\\
1-X&1-Y&-1&-1&0&0&0\\\\
1-X&1-Y&-1&-1&0&0&0\\\\
\end{matrix}  \right]  \tag{1} $$


XY11100:
$$\left[  \begin{matrix}
0&X-Y&X-1&X-1&X-1&X&X\\\\
Y-X&0&Y-1&Y-1&Y-1&Y&Y\\\\
1-X&1-Y&0&0&0&1&1\\\\
1-X&1-Y&0&0&0&1&1\\\\
1-X&1-Y&0&0&0&1&1\\\\
-X&-Y&-1&-1&-1&0&0\\\\
-X&-Y&-1&-1&-1&0&0\\\\
\end{matrix}  \right]  \tag{2}$$
XY00111YX:
$$\left[  \begin{matrix}
0&X-Y&X&X&X-1&X-1&X-1&X-Y&0\\\\
Y-X&0&Y&Y&Y-1&Y-1&Y-1&0&Y-X\\\\
-X&-Y&0&0&-1&-1&-1&-Y&-X\\\\
-X&-Y&0&0&-1&-1&-1&-Y&-X\\\\
1-X&1-Y&-1&-1&0&0&0&1-Y&1-X\\\\
1-X&1-Y&-1&-1&0&0&0&1-Y&1-X\\\\
1-X&1-Y&-1&-1&0&0&0&1-Y&1-X\\\\
Y-X&0&Y&Y&Y-1&Y-1&Y-1&0&Y-X\\\\
0&X-Y&X&X&X-1&X-1&X-1&X-Y&0\\\\
\end{matrix}  \right]  \tag{3}$$

XY11100YX:
$$\left[  \begin{matrix}
0&X-Y&X-1&X-1&X-1&X&X&X-Y&0\\\\
Y-X&0&Y-1&Y-1&Y-1&Y&Y&0&Y-X\\\\
1-X&1-Y&0&0&0&1&1&1-Y&1-X\\\\
1-X&1-Y&0&0&0&1&1&1-Y&1-X\\\\
1-X&1-Y&0&0&0&1&1&1-Y&1-X\\\\
-X&-Y&-1&-1&-1&0&0&-Y&-X\\\\
-X&-Y&-1&-1&-1&0&0&-Y&-X\\\\
Y-X&0&Y-1&Y-1&Y-1&Y&Y&0&Y-X\\\\
0&X-Y&X-1&X-1&X-1&X&X&X-Y&0\\\\
\end{matrix}  \right]  \tag{4}$$

As we can see, for the difference matrices with channel-reference concatenated at both sides (i.e., XY00111YX, XY11100YX), they present the very similar pattern, that is, matrix (4) is exactly the result by rotating the matrix (3) clockwise 180 degrees. Since we encode the difference matrix using CNNs, which have translation invariance and rotation invariance[1]. Thus these two matrices would be infered as the same property by CNNs, leading to a more generalized seizure pattern. In contrast, the former two segments (XY00111, XY11100) without reversed right channel-reference (R_cl) involved, their difference matrices (1,2) cannot be obtained from each other by translation or rotation, which increases the burden of model learning.

**GR2. Clarify the data division and the hyperparameters selection.**

**Data Division.** We conducted experiments on one clinical dataset and two public datasets. To perform experiments in a domain generalization setting, we grouped subjects within each dataset and then constructed different folds. Each fold comprised a training set, a validation set, and a test set, each consisting of one or multiple groups. For detailed experiment setup, please refer to Appendix-D.

**Hyperparameter Selection.** Our hyperparameter selection was not based on results from the test set. The model underwent evaluation using Leave-one-domain-out validation [2] with grid search. We opted for hyperparameters that maximize the average F1 score across the held-out domains in all folds. The hyperparameter search scope is as follows.

| Hyperparameter     | Search Scope                                                 |
| ------------------ | ------------------------------------------------------------ |
| Length of Segment  | Clinical:{100, 150, 200, 250,300} Public:{300, 400, 500, 600, 700} |
| Number of Segment  | {8, 9, 10, 11, 12}                                           |
| Number of Cluster  | {6, 8, 10, 12}                                               |
| Base filter Number | {8, 16, 32, 64, 128}                                         |
| Learning Rate      | {1e-4, 3e-4, 5e-4, 1e-3, 3e-3}                               |
| Batch Size         | {8, 24, 32, 40}                                              |

**GR3. Discuss the empirical and theoretical supports of DMNet.**

Basically, the design of our proposed DMNet is motivated by both neuroscience discoveries and our empirical analyses.

**Neuroscience Basis.** Although there is domain shift among different subjects and even within the same subject at different times, seizure events consistently show a higher average amplitude in the frequency domain compared to their background signal. This discovery aligns with previous research in the field [3].

**Empirical Analysis.** Building on this, we propose a self-comparison mechanism that compares the target segment with its adjacent normal segment to reduce domain shifts between subjects and time intervals. Our preliminary analysis in Section 3 reveals that using a subtraction operation in the frequency domain for self-comparison effectively reduces domain shifts while enhancing the discriminability of seizures from normal events. The success of this self-comparison method may stem from the fact that subtraction-based comparison is a relative concept, capable of mitigating data scale discrepancies while explicitly highlighting the differences between seizure and normal events.

We appreciate reviewers' constructive comments regarding the lack of theoretical support. Up to the present, why the difference matrix is able to reduce the distribution shift between subjects remains an open question, and we'll leave as our future work.

[1]Zeiler, el.. Visualizing and Understanding Convolutional Networks.

[2]Gulrajani I, el. In Search of Lost Domain Generalization[C] International Conference on Learning Representations.

[3]Catherine el. Evidence of an inhibitory restraint of seizure activity in humans. Nature communications, 2012.

---

> ### Author Response · Authors · 2024-08-08
> **Some minor typos fixes in GR1**
>
> **Some minor typos fixes in GR1**
>
> I apologize for the typo in formula 1 and 3 in GR1. Below is the corrected formula 1 and 3.
>
> XY00111:
> $$
> \left[  \begin{matrix}
> 0&X-Y&X&X&X-1&X-1&X-1\\\\
> Y-X&0&Y&Y&Y-1&Y-1&Y-1\\\\
> -X&-Y&0&0&-1&-1&-1\\\\
> -X&-Y&0&0&-1&-1&-1\\\\
> 1-X&1-Y&1&1&0&0&0\\\\
> 1-X&1-Y&1&1&0&0&0\\\\
> 1-X&1-Y&1&1&0&0&0\\\\
> \end{matrix}  \right]  \tag{1}
> $$
>
>
> XY00111YX:
> $$
> \left[  \begin{matrix}
> 0&X-Y&X&X&X-1&X-1&X-1&X-Y&0\\\\
> Y-X&0&Y&Y&Y-1&Y-1&Y-1&0&Y-X\\\\
> -X&-Y&0&0&-1&-1&-1&-Y&-X\\\\
> -X&-Y&0&0&-1&-1&-1&-Y&-X\\\\
> 1-X&1-Y&1&1&0&0&0&1-Y&1-X\\\\
> 1-X&1-Y&1&1&0&0&0&1-Y&1-X\\\\
> 1-X&1-Y&1&1&0&0&0&1-Y&1-X\\\\
> Y-X&0&Y&Y&Y-1&Y-1&Y-1&0&Y-X\\\\
> 0&X-Y&X&X&X-1&X-1&X-1&X-Y&0\\\\
> \end{matrix}  \right]  \tag{3}
> $$

---

> ### Comment · Reviewer_EpQU · 2024-08-11
> **Hyper-parameter selection confusion**
>
> Thank you for the clarification on the reason to add channel-level reference on both sides, it dose makes sense due to CNNs local feature extraction. However, I disagree about CNNs properties. CNNs are not completely shift and rotation invariant, but can exhibit some local shift invariance through pooling, and rotation invariance can be also learned but is not intrinsic.
>
>
> The hyper-parameter selection via leave-one-domain out (LODO) generalization [2] is more involved than what is conducted in the additional experiments in the rebuttal. The rebuttal states "Maximize the average F1 across the held-out domains in all folds". But the PDF says only 1 dataset was used as source. Does this mean the domains are subjects like in the main body?  Otherwise, if dataset are domains to do the process of LODO, one needs at least two domains as source. Hyper-parameters that perform good on average (across either held out folds of subjects or held out domains) are then tested on the unused target domains.
>
> Could the authors report the selected hyper-parameters that were selected for each test set (by "for" I mean that they were selected without seeing the test set, but were selected with the intent of use on this test set)?
>
> Similarly, I'm still not convinced that a rigorous selection was used in the main body results for the 2-1-1 or 4-1-1 strategy.  There should be a different set of hyper-parameters based on the  validation set in each fold, rather than one global one. Choosing one set of hyper-parameters based on average performance (without an additional test set) is not rigorous.
>
>
> Table 3 in PDF must have wrong column labels for target domains...

---

> ### Author Response · Authors · 2024-08-12
> **Clarification of Hyper-parameter selection**
>
> Please refer to our latest responses to the questions posed by **Reviewer EpQU**.
>
> **Minors:**
>
> Thank you for pointing out the typo in Table 3 of the attached PDF.
>
> The "FNUSA" in Table 3 should be corrected to "Clinical," and I will make the necessary correction in the manuscript.

---

### Decision · Program_Chairs · 2024-09-25

**Decision:**

Accept (poster)

**Comment:**

The authors present Difference Matrix-based Neural Network for subject-independent automatic seizure detection. The model addresses the domain shift in iEEG signals across different subjects by leveraging a self-comparison mechanism that can enhance the model's ability to perform effectively across new subjects without additional training. The approach is novel and well presented. Several concerns on empirical evaluation and hyper-parameter selection process are well addressed by the additional experiments and explanations in the authors' rebuttal.